# MetaP: How to Transfer Your Knowledge on Learning Hidden Physics

## Abstract

Gradient-based meta-learning methods have primarily focused on classical machine learning tasks such as image classification and function regression, where they were found to perform well by recovering the underlying common representation among a set of given tasks. Recently, PDE-solving deep learning methods, such as neural operators, are starting to make an important impact on learning and predicting the response of a complex physical system directly from observational data. Since the data acquisition in this context is commonly challenging and costly, the call of utilization and transfer of existing knowledge to new and unseen physical systems is even more acute.

Herein, we propose a novel meta-learnt approach for transfer-learning knowledge between neural operators, which can be seen as transferring the knowledge of solution operators between governing (unknown) PDEs with varying parameter fields. With the key theoretical observation that the underlying parameter field can be captured in the first layer of the neural operator model, in contrast to typical final-layer transfer in existing meta-learning methods, our approach is a provably universal solution operator for multiple PDE solving tasks. As applications, we demonstrate the efficacy of our proposed approach on PDE-based datasets and a real-world material modeling problem, demonstrating that our method can handle complex and nonlinear physical response learning tasks while greatly improving the sampling efficiency in new and unseen tasks.

## 1 Introduction

Few-shot learning is an important problem in machine learning, where new tasks are learned with a very limited number of labelled datapoints (Wang et al., 2020). In recent years, significant progress has been made on few-shot learning using meta-learning approaches (Koch et al., 2015; Vinyals et al., 2016; Snell et al., 2017; Finn et al., 2017; Santoro et al., 2016; Antoniou et al., 2018; Ravi & Larochelle, 2016; Nichol & Schulman, 2018; Raghu et al., 2019; Tripuraneni et al., 2021; Collins et al., 2022). Broadly speaking, given a family of tasks, some of which are used for training and others for testing, meta-learning approaches aim to learn a shared multi-task representation that can generalize across the different training tasks, and result in fast adaptation to new and unseen testing tasks. Although most of meta-learning learning developments focus on conventional machine learning problems such as image classification, function regression, and reinforcement learning, studies on few-shot learning approaches for complex physical system modeling problems have been limited. The call of developing a few-shot learning approach for complex physical system modeling problems is just as acute, while the typical understanding of how multi-task learning should be applied on this scenario is still nascent.

As a motivating example, we consider the scenario of new material discovery in the lab environment, where the material model is built based on experimental measurements of its responses subject to different loadings. Since the physical properties (such as the mechanical and structural parameters) in different material specimens vary, the model learnt from experimental measurements on one specimen would have a large generalization error on future specimens. That means, the data-driven model has to be trained repeatedly with a large number of material specimens, which makes the learning process inefficient. Further, experimental measurement acquisition of these specimens is often challenging and expensive. In some problems, a large amount of measurements are not even

feasible. For example, in the design and testing of biosynthetic tissues, performing repeated loading would potentially induces the cross-linking and permanent set phenomenon, which notoriously alter the tissue durability (Zhang & Sacks, 2017). As a result, it is critical to learn the physical response model of a new specimen with samples sizes as small as possible. Furthermore, since many characterization methods to obtain underlying material mechanistic and structural properties would require the use of destructive methods (Misfeld & Sievers, 2007; Rieppo et al., 2008), in practice many physical properties are not measured and can only be treated as hidden and unknown variables. We likely only have limited access to the measurements on the complex system responses caused by the change of these physical properties.

Supervised operator learning methods are typically used to address this class of problems. They take a number of observations on the loading field as input, and try to predict the corresponding physical system response field as output, corresponding to one underlying PDE (as one task). Herein, we consider the meta-learning of multiple complex physical systems (as tasks), such that all these tasks are governed by a common PDE with different (hidden) physical property or parameter fields. Formally, assume that we have a distribution $p(\mathcal{T})$ over tasks, each task $\mathcal{T}^\eta \sim p(\mathcal{T})$ corresponds to a hidden physical property field $\mathbf{b}^\eta(\mathbf{x}) \in \mathcal{B}(\mathbb{R}^{d_b})$ that contains the task-specific mechanistic and structural information in our material modeling example. On task $\mathcal{T}^\eta$, we have a number of observations on the loading field $\mathbf{g}_i^\eta(\mathbf{x}) \in \mathcal{A}(\mathbb{R}^{d_g})$ and the corresponding physical system response field $\mathbf{u}_i^\eta(\mathbf{x}) \in \mathcal{U}(\mathbb{R}^{d_u})$ according to a hidden parameter field $\mathbf{b}^\eta(\mathbf{x})$. Here, $i$ is the sample index, $\mathcal{B}$, $\mathcal{A}$ and $\mathcal{U}$ are Banach spaces of function taking values in $\mathbb{R}^{d_b}$, $\mathbb{R}^{d_g}$ and $\mathbb{R}^{d_u}$, respectively. For task $\mathcal{T}^\eta$, our modeling goal is to learn the solution operator $\mathcal{G}^\eta : \mathcal{A} \to \mathcal{U}$, such that the learnt model can predict the corresponding physical response field $\mathbf{u}(\mathbf{x})$ for any loading field $\mathbf{g}(\mathbf{x})$. Without transfer learning, one needs to learn a surrogate solution operator for each task only based on the data pairs on this task, and repeat the training for every task. The learning procedure would require a relatively large amount of observation pairs and training time for each task. Therefore, this physical-based modeling scenario raises a key question: *Given knowledge on a number of parametric PDE solving tasks with different unknown parameters, how can one efficiently learn the best surrogate solution operator for a new and unknown parameter, with only a small set of training data pairs*[1]*?*

To address this question, we introduce MetaP, a novel meta-learnt approach for transfer-learning knowledge between neural operators, which can be seen as transferring the knowledge of solution operators between governing (unknown) PDEs with varying hidden parameter fields. Our **main contributions** are:

- MetaP is the first neural-operator-based approach for multi-task learning, which not only preserves the generalizability to different resolutions and input functions from its integral neural operator architecture, but also improves sampling efficiency on new tasks – for comparable accuracy, MetaP saves the required number of measurements by $\sim$90%.

- With rigorous operator approximation analysis, we made the key observation that the hidden parameter field can be captured by adapting the first layer of the neural operator model, in contrast to typical final-layer transfer in existing meta-learning methods. By construction, MetaP serves as a provably universal solution operator for multiple PDE solving tasks.

- From synthetic, benchmark, to a real-world biological tissue datasets, the proposed method consistently outperforms existing baseline gradient-based meta-learning methods.

## 2 BACKGROUND AND RELATED WORK

In this section we introduce the relevant materials on hidden physics learning, neural operators, and gradient-based meta-learning methods, which will later complement the definition of our method.

### 2.1 HIDDEN PHYSICS LEARNING AND NEURAL OPERATORS

For many decades, physics-based PDEs have been commonly employed for predicting and monitoring complex system responses, then traditional numerical methods were employed to solve these

---

[1]In some meta-learning literature (e.g., (Xu et al., 2020)), these small sets of labelled data pairs on a new task (or any task) is also called the context, and the learnt model will be evaluated on an additional set of unlabelled data pairs, i.e., the target.

PDEs and provide predictions for desired system responses. However, three fundamental challenges present. First, the choice of governing PDE laws is often determined *a priori* and free parameters are often tuned to obtain agreement with experimental data. This fact makes the rigorous calibration and validation process challenging. Second, traditional numerical methods are solved for specific boundary and initial conditions, as well as loading or source terms. Therefore, they are not generalizable for other operating conditions and hence not effective for real-time prediction. Third, complex PDE systems such as turbulence flows and heterogeneous materials modeling problems usually require a very fine discretization, and therefore very time-consuming for traditional solvers.

To provide an efficient surrogate model for physical responses, machine learning methods may hold the key. Recently, there has been significant progress in the development of deep neural networks (NNs), focusing on learning the hidden physics of a complex system (Ghaboussi et al., 1998; 1991; Carleo et al., 2019; Karniadakis et al., 2021; Zhang et al., 2018; Cai et al., 2022; Pfau et al., 2020; He et al., 2021; Besnard et al., 2006). Among these methods, the neural operators show particular promises in resolving the above challenges. Neural operators aim to learn maps between inputs of a dynamical system and its state, so that the network can serve as a surrogate for a solution operator (Li et al., 2020a;b;c; You et al., 2022a; Ong et al., 2022; Gupta et al., 2021; Lu et al., 2019; 2021b; Goswami et al., 2022a).

Comparing with the classical NNs, the most notable advantages of neural operators are resolution independence and generalizability to different input instances. Moreover, comparing with the classical PDE modeling approaches, neural operators require only data with no knowledge of the underlying PDE. All these advantages make neural operators promising tools to PDE learning tasks. Examples include modeling the unknown physics law of real-world problems (Yin et al., 2022a; Goswami et al., 2022a; Yin et al., 2022b), and providing efficient solution operator for PDEs (Li et al., 2020a;b;c; Lu et al., 2021c;a). On the other hand, data in scientific applications are often scarce and incomplete. Utilization of other relevant data sources could alleviate such a problem, yet no existing work have addressed the transferability of neural operators. Through the meta-learning techniques, our work fulfills the demand of such a transfer setting, with the same type of PDE system but different (hidden) physical properties.

## 2.2 GRADIENT-BASED META-LEARNING METHODS

One highly successful meta-learning algorithm has been Model Agnostic Meta-Learning (MAML) (Finn et al., 2017), which led to the development of a series of related gradient-based meta-learning (GBML) methods (Raghu et al., 2019; Nichol & Schulman, 2018; Antoniou et al., 2018; Hospedales et al., 2020). Almost-No-Inner-Loop algorithm (ANIL) (Raghu et al., 2019) modifies MAML by freezing the final layer representation during local adaptation. Recently, theoretical analysis (Collins et al., 2022) found that the driving force causing MAML and ANIL to recover the general representation is the adaptation of the final layer of their models, which harnesses the underlying task diversity to improve the representation in all directions of interest.

Although MAML and the general meta-learning approaches have achieved impressive performance in some machine-learning applications such as the image classification and reinforcement learning scenarios, a few work has studied the hidden physics learning under meta (Mai et al., 2021; Zhang et al., 2022; Yin et al., 2021; Wang et al., 2021) or even transfer setting (Kailkhura et al., 2019; Goswami et al., 2022b). Among these meta-learning works, (Mai et al., 2021; Zhang et al., 2022) are designed for specific physical applications, while (Yin et al., 2021; Wang et al., 2021) focus on on dynamics forecasting by learning the temporal evolution information directly (Yin et al., 2021) or learning time-invariant features (Wang et al., 2021). Hence, none of these works have provided a generic approach nor theoretical understanding on how to transfer the multi-task knowledge between a series of complex physical systems, such that all these tasks are governed by a common parametric PDE with different physical parameters.

## 3 META-LEARNT NEURAL OPERATOR

### 3.1 INTEGRAL NEURAL OPERATORS

Here, we first state the base model of this work without the meta aspect. The integral neural operators, first proposed in (Li et al., 2020a) and further developed in (Li et al., 2020b;c; You et al.,

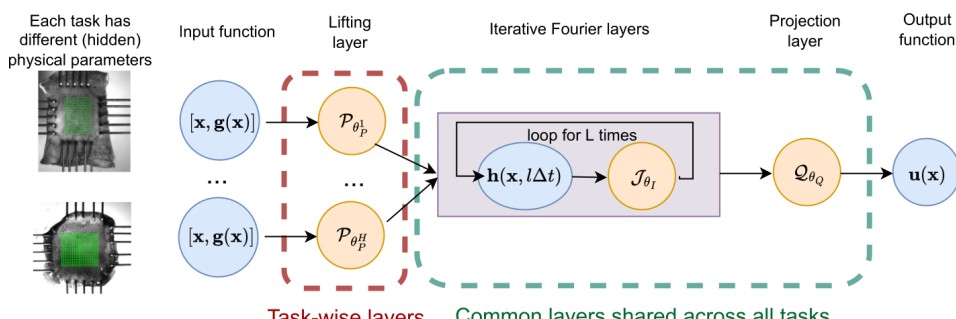

Figure 1: The architecture of MetaP based on an integral neural operator model.

2022a;c) comprises of three building blocks. First, the input function, $\mathbf{g}(\mathbf{x}) \in \mathcal{A}$, is lifted to a higher dimensional representation via $\mathbf{h}(\mathbf{x}, 0) = \mathcal{P}[\mathbf{g}](\mathbf{x}) := P(\mathbf{x})[\mathbf{x}, \mathbf{g}(\mathbf{x})]^T + \mathbf{p}(\mathbf{x})$. Here, $P(\mathbf{x}) \in \mathbb{R}^{(s+d_g) \times d_h}$ and $\mathbf{p}(\mathbf{x}) \in \mathbb{R}^{d_h}$ define an affine pointwise mapping, which are often taken as constant parameters, i.e., $P(\mathbf{x}) \equiv P$ and $\mathbf{p}(\mathbf{x}) \equiv \mathbf{p}$. Then, the feature vector function $\mathbf{h}(\mathbf{x}, 0)$ goes through an iterative layer block where the layer update is defined via the action of the sum of a local linear operator, a nonlocal integral kernel operator, and a bias function: $\mathbf{h}(\cdot, (l+1)\Delta t) = \mathcal{J}_{l+1}[\mathbf{h}(\cdot, l\Delta t)]$, for $l = 0, \cdots, L-1$. Here, $\mathbf{h}(\cdot, j\Delta t), j = 0, \cdots, L := T/\Delta t$, is a sequence of functions representing the values of the network at each hidden layer, taking values in $\mathbb{R}^{d_h}$. $\mathcal{J}_1, \cdots, \mathcal{J}_L$ are the nonlinear operator layers, defined by the particular choice of networks. In this work, we employ the implicit Fourier neural operator (IFNO) as the base model, because of its theoretical universal approximation property in PDE solving tasks (You et al., 2022c) and robustness in complex physical response modeling tasks (You et al., 2022b)[2]. In this case, the iterative layers are taken as $\mathcal{J}_1 = \cdots = \mathcal{J}_L = \mathcal{J}$, where

$$\mathbf{h}(\mathbf{x}, (l+1)\Delta t) = \mathcal{J}[\mathbf{h}(\mathbf{x}, l\Delta t)]$$
$$:= \mathbf{h}(\mathbf{x}, l\Delta t) + \Delta t \sigma \left( W \mathbf{h}(\mathbf{x}, l\Delta t) + \mathcal{F}^{-1}[\mathcal{F}[\kappa(\cdot; \mathbf{v})] \cdot \mathcal{F}[\mathbf{h}(\cdot, l\Delta t)]](\mathbf{x}) + \mathbf{c}(\mathbf{x}) \right). \quad (1)$$

Here, $\mathcal{F}$ and $\mathcal{F}^{-1}$ denote the Fourier transform and its inverse, respectively. In practice, $\mathcal{F}$ and $\mathcal{F}^{-1}$ are computed using the FFT and its inverse to each component of $\mathbf{h}$ separately. Also, $\mathbf{c} \in \mathbb{R}^{d_h}$ defines a constant bias, $W \in \mathbb{R}^{d_h \times d_h}$ is the weight matrix, and $\mathcal{F}[\kappa(\cdot; \mathbf{v})] := R$ is a circulant matrix that depends on the convolution kernel $\kappa$. $\sigma$ is an activation function, which is oftenly taken to be the popular rectified linear unit (ReLU) function. Finally, the output $\mathbf{u}(\cdot) \in \mathcal{U}$ is obtained through a projection layer. In particular, we project the last hidden layer representation $\mathbf{h}(\cdot, T)$ onto $\mathcal{U}$ as: $\mathbf{u}(\mathbf{x}) = \mathcal{Q}[\mathbf{h}(\cdot, T)](\mathbf{x}) := Q_2(\mathbf{x})\sigma(Q_1\mathbf{h}(\mathbf{x}, T) + \mathbf{q}_1(\mathbf{x})) + \mathbf{q}_2(\mathbf{x})$. Here, $Q_1(\mathbf{x}) \in \mathbb{R}^{d_Q \times d_h}$, $Q_2(\mathbf{x}) \in \mathbb{R}^{d_u \times d_Q}$, $\mathbf{q}_1(\mathbf{x}) \in \mathbb{R}^{d_Q}$ and $\mathbf{q}_2(\mathbf{x}) \in \mathbb{R}^{d_u}$ are the appropriately sized matrices and vectors that are part of the parameter set that we aim to learn. Similarly as for the lifting layer, $Q_1(\mathbf{x})$, $Q_2(\mathbf{x})$, $\mathbf{q}_1(\mathbf{x})$ and $\mathbf{q}_2(\mathbf{x})$ are also often taken as constant parameters, which will be denoted as $Q_1$, $Q_2$, $\mathbf{q}_1$ and $\mathbf{q}_2$, respectively. In the following, we denote the set of trainable parameters in the lifting layer as $\theta_P$, the set from the iterative layer block as $\theta_I$, and the set in the projection layer as $\theta_Q$.

The neural operator can be employed to learn an approximation for the solution operator, $\mathcal{G}$. Given $\mathcal{D} := \{(\mathbf{g}_i, \mathbf{u}_i)\}_{i=1}^N$, a labelled (context) set of observations, where the input $\{\mathbf{g}_i\} \subset \mathcal{A}$ is a set of independent and identically distributed (i.i.d.) random fields from a known probability distribution $\mu$ on $\mathcal{A}$, and $\mathbf{u}_i(\mathbf{x}) \in \mathcal{U}$, possibly noisy, is the observed corresponding solution, let $\Omega \subset \mathbb{R}^s$ be the domain of interests, we assume that all observations can be modeled with a parametric PDE form

$$\mathcal{K}_{\mathbf{b}(\mathbf{x})}[\mathbf{u}_i](\mathbf{x}) = \mathbf{g}_i(\mathbf{x}), \quad \mathbf{x} \in \Omega. \quad (2)$$

Here, $\mathcal{K}_{\mathbf{b}}$ is the operator representing the possibly unknown governing law, e.g., balance laws. Then, the system response can be learnt by constructing a surrogate solution operator of equation 2: $\tilde{\mathcal{G}}[\mathbf{g}; \theta](\mathbf{x}) := \mathcal{Q}_{\theta_Q} \circ (\mathcal{J}_{\theta_I})^L \circ \mathcal{P}_{\theta_P}[\mathbf{g}](\mathbf{x}) \approx \mathbf{u}(\mathbf{x})$, where the parameter set $\theta = [\theta_P, \theta_I, \theta_Q]$ is obtained by solving the optimization problem:

$$\min_{\theta \in \Theta} \mathcal{L}_{\mathcal{D}}(\theta) = \min_{\theta \in \Theta} \mathbb{E}_{\mathbf{f} \sim \mu}[C(\tilde{\mathcal{G}}[\mathbf{g}; \theta], \mathcal{G}[\mathbf{g}])] \approx \min_{\theta \in \Theta} \sum_{i=1}^N [C(\tilde{\mathcal{G}}[\mathbf{g}_i; \theta], \mathbf{u}_i)]. \quad (3)$$

Here $C$ denotes a properly defined cost functional which is often taken as the the mean square error.

---

[2]We also point out that the proposed multi-task strategy is generic and hence also applicable to any other integral neural operators (Li et al., 2020a;b;c; You et al., 2022a).

## 3.2 BASE META MODEL WITH MAML AND ANIL

To transfer the multi-task knowledge between a series of complex systems governed by different hidden physical parameters, we proposed to leverage the integral neural operator with a meta-learning setting. Herein, assume that for each training task $\mathcal{T}^\eta \sim p(\mathcal{T})$ we have a set of observations of loading field/respond field data pairs $\mathcal{D}^\eta := \{(\mathbf{g}_i^\eta(\mathbf{x}), \mathbf{u}_i^\eta(\mathbf{x}))\}_{i=1}^{N^\eta}$, and each task can be modeled with a parametric PDE form

$$\mathcal{K}_{\mathbf{b}^\eta(\mathbf{x})}[\mathbf{u}_i^\eta](\mathbf{x}) = \mathbf{g}_i^\eta(\mathbf{x}), \quad \mathbf{x} \in \Omega, \tag{4}$$

where $\mathbf{b}^\eta(\mathbf{x})$ is the hidden task-specific physical parameter field for the common governing law. Given a new and unseen test task, $\mathcal{T}^{test}$, and a context set of labelled samples $\mathcal{D}^{test} := \{(\mathbf{g}_i^{test}(\mathbf{x}), \mathbf{u}_i^{test}(\mathbf{x}))\}_{i=1}^{N^{test}}$ on it, our goal is to obtain the approximated solution operator model on the test task as $\tilde{\mathcal{G}}[\mathbf{g}; \theta^{test}]$.

A straightforward approach would be to simply apply MAML and ANIL to a neural operator architecture, which will be treated as the baselines of our studies. Here we formally state our implementation of ANIL and MAML for the problem described above.

**MAML.** The MAML algorithm proposed in (Finn et al., 2017) aims to find an initialization, $\tilde{\theta}$, across all tasks, so that new tasks can be learnt with very few examples. First, we draw a batch $\{\mathcal{T}^\eta\}_{\eta=1}^H$ of $H$ tasks from $p(\mathcal{T})$. For each task $\mathcal{T}^\eta$, we split the available set of loading field/response field data pairs $\mathcal{D}^\eta$ to a support set of samples, $\mathcal{S}^\eta$, which will be used for inner loop updates, and a target set of samples, $\mathcal{Z}^\eta$, for outer loop updates. Then, for the inner loop we let $\theta^{\eta,0} := \tilde{\theta}$ and $\theta^{\eta,i}$ be the task-wise parameter after $i$ gradient updates. During each inner loop update, we compute

$$\theta^{\eta,i} = \theta^{\eta,i-1} - \alpha \nabla_{\theta^{\eta,i-1}} \mathcal{L}_{\mathcal{S}^\eta}(\theta^{\eta,i-1}), \quad \text{for } \eta = 1, \cdots, H, \tag{5}$$

where $\mathcal{L}_{\mathcal{S}^\eta}(\theta^{\eta,i-1})$ is the loss on the support set of the $\eta$-th task, and $\alpha$ is the step size. After $m$ inner loop updates, we update the initial parameter $\tilde{\theta}$ with a fixed step size $\beta$:

$$\tilde{\theta} \leftarrow \tilde{\theta} - \beta \nabla_{\tilde{\theta}} \mathcal{L}_{\text{meta}}(\tilde{\theta}), \text{ where the meta-loss } \mathcal{L}_{\text{meta}}(\tilde{\theta}) := \sum_{\eta=1}^H \mathcal{L}_{\mathcal{Z}^\eta}(\theta^{\eta,m}). \tag{6}$$

Then, on the test task, $\mathcal{T}^{test}$, we perform inner loop adaptation based on few labelled samples $\mathcal{D}^{test}$ until convergence, and obtain the approximated solution operator model on the test task as $\tilde{\mathcal{G}}[\mathbf{g}; \theta^{test}]$.

**ANIL.** In (Raghu et al., 2019), ANIL was proposed as a modified version of MAML with inner loop updates only for the final layer. The inner loop update formulation equation 5 is modified as

$$\theta_Q^{\eta,i} = \theta_Q^{\eta,i-1} - \alpha \nabla_{\theta_Q^{\eta,i-1}} \mathcal{L}_{\mathcal{S}^\eta}(\theta_Q^{\eta,i-1}), \quad \text{for } \eta = 1, \cdots, H, \tag{7}$$

where $\theta_Q^{\eta,i}$ is the task-wise parameter on the projection layer after $i$ gradient updates. Then, we perform the same outer loop updates following equation 6.

## 3.3 METAP: A NOVEL META-LEARNT NEURAL OPERATOR ARCHITECTURE

We now propose MetaP, which *applies task-wise adaptation only to the first layer, i.e., the lifting layer*, with the full algorithm outlined in Algorithm 1. Similar as in other meta-learning approaches (Yoon et al., 2018; Vanschoren, 2018; Yang & Kwok, 2022; Kalais & Chatzis, 2022), the algorithm consists of two phases: (1) a meta-train phase which learns shared iterative layers parameters $\theta_I$ and projection layer parameters $\theta_P$ from existing tasks; (2) a meta-test phase which transfers the learned knowledge and rapidly learning surrogate solution operators for unseen tasks with unknown physical parameter field, where only a few test samples are required.

To see the inspiration of the proposed architecture, without loss of generality, we assume that the underlying task parameter field $\mathbf{b}^\eta(\mathbf{x})$, modeling the physical property field, is normalized and satisfying $\left\| \mathbf{b}^\eta(\mathbf{x}) - \overline{\mathbf{b}}(\mathbf{x}) \right\|_{L^2(\Omega)} \leq 1$ for all $\eta \in \{1, \cdots, H\}$, where $\overline{\mathbf{b}} := \mathbb{E}_{\mathcal{T}^\eta \sim p(\mathcal{T})}[\mathbf{b}^\eta]$. Denoting $\mathcal{F}_{\mathbf{u}}[\mathbf{b}] := \mathcal{K}_{\mathbf{b}}[\mathbf{u}]$ as a function from physical parameter fields $\mathcal{B}$ to loading fields $\mathcal{A}$, we can take the Fréchet derivative of $\mathcal{F}$ with respect to $\mathbf{b} - \overline{\mathbf{b}}$ and obtain:

$$\mathcal{K}_{\mathbf{b}^\eta}[\mathbf{u}] = \mathcal{F}_{\mathbf{u}}[\overline{\mathbf{b}}] + D\mathcal{F}_{\mathbf{u}}[\overline{\mathbf{b}}](\mathbf{b}^\eta - \overline{\mathbf{b}}) + o(\left\| \mathbf{b}^\eta - \overline{\mathbf{b}} \right\|_{L^2(\Omega)}).$$

---

**Meta-Train Phase:**
**Require:** a batch $\{\mathcal{T}^\eta\}_{\eta=1}^H$ of known tasks and available data pairs $\mathcal{D}^\eta := \{(\mathbf{g}_i^\eta(\mathbf{x}), \mathbf{u}_i^\eta(\mathbf{x}))\}_{i=1}^{N^\eta}$ on each task
**Output:** common parameters $\theta_I^*$ and $\theta_Q^*$ across all tasks
1. randomly initialize $\theta_I$, $\theta_Q$, and $\{\theta_P^\eta\}_{\eta=1}^H$
2. solve the optimization problem:

$$\{\theta_I^*, \theta_Q^*, \{\theta_P^{\eta,*}\}_{\eta=1}^H\} = \operatorname*{argmin}_{\{\theta_I, \theta_Q, \{\theta_P^\eta\}_{\eta=1}^H\}} \sum_{\eta=1}^H \mathcal{L}_{\mathcal{D}^\eta}([\theta_P^\eta, \theta_I, \theta_Q])$$

---

**Meta-Test Phase:**
**Require:** a test task $\mathcal{T}^{\text{test}}$ and few labelled data pairs $\mathcal{D}^{\text{test}} := \{(\mathbf{g}_i^{\text{test}}(\mathbf{x}), \mathbf{u}_i^{\text{test}}(\mathbf{x}))\}_{i=1}^{N^{\text{test}}}$
**Output:** the task-wise parameter $\theta_P^{\text{test},*}$ and the corresponding surrogate PDE solution operator $\tilde{\mathcal{G}}[\mathbf{g}; [\theta_P^{\text{test},*}, \theta_I^*, \theta_Q^*]](\mathbf{x})$ for the test task
3. solve for the lift layer parameter from the optimization problem:

$$\theta_P^{\text{test},*} = \operatorname*{argmin}_{\theta_P^{\text{test}}} \mathcal{L}_{\mathcal{D}^{\text{test}}}([\theta_P^{\text{test}}, \theta_I^*, \theta_Q^*])$$

**Algorithm 1:** MetaP for Few-Shot Learning of New PDE Solver with Hidden Physical Parameters

Substituting the above formulation into equation 4, we obtain

$$\mathcal{F}_{\mathbf{u}_i^\eta}[\overline{\mathbf{b}}] + D\mathcal{F}_{\mathbf{u}_i^\eta}[\overline{\mathbf{b}}](\mathbf{b}^\eta - \overline{\mathbf{b}}) \approx \mathbf{g}_i^\eta.$$

Denoting $\mathbf{F}_1[\mathbf{b}^\eta] := [\mathbf{1}, \mathbf{b}^\eta - \overline{\mathbf{b}}]$ and $\mathbf{F}_2[\mathbf{u}_i^\eta] := [\mathcal{F}_{\mathbf{u}_i^\eta}[\overline{\mathbf{b}}], D\mathcal{F}_{\mathbf{u}_i^\eta}[\overline{\mathbf{b}}]]$, we can actually reformulated equation 4 into a more generic form:

$$\mathbf{F}_1[\mathbf{b}^\eta](\mathbf{x}) \cdot \mathbf{F}_2[\mathbf{u}_i^\eta](\mathbf{x}) = \mathbf{g}_i^\eta(\mathbf{x}), \quad \mathbf{x} \in \Omega. \tag{8}$$

We point out that this parametric PDE form is indeed very general and finds applications in many science and engineering applications – besides our motivating example on material modeling, examples also include the monitoring of tissue degeneration problems (Zhang & Sacks, 2017), the detection of subsurface flows (Dejam et al., 2017), the nondestructive inspection in aviation (Fallah et al., 2019), and the prediction of concrete structures deterioration (Wei et al., 2021), etc.

In the following, we show that MetaP are universal solution finding operators for the multi-task PDE solving problem in equation 8, in the sense that they can approximate a fixed point method to a desired accuracy. For simplicity, we consider a $1D$ domain $\Omega$, and scalar functions $\mathbf{F}_1[\mathbf{b}^\eta]$, $\mathbf{F}_2[\mathbf{u}_i^\eta]$. These functions are assumed to be sufficiently smooth and measured at uniformly distributed nodes $\chi := \{\mathbf{x}_1, \mathbf{x}_2, \ldots, \mathbf{x}_M\}$, with $\mathbf{F}_1[\mathbf{b}^\eta](\mathbf{x}_j) \neq 0$ for all $\eta$ and $j$. Then, equation 8 can be formulated as an implicit system of equations:

$$\mathcal{H}(\mathbf{U}_i^{\eta,*}; \tilde{\mathbf{G}}_i^\eta) := \begin{bmatrix} \mathbf{F}_2[\mathbf{u}_i^\eta](\mathbf{x}_1) - \mathbf{g}_i^\eta(\mathbf{x}_1)/\mathbf{F}_1[\mathbf{b}^\eta](\mathbf{x}_1) \\ \vdots \\ \mathbf{F}_2[\mathbf{u}_i^\eta](\mathbf{x}_M) - \mathbf{g}_i^\eta(\mathbf{x}_M)/\mathbf{F}_1[\mathbf{b}^\eta](\mathbf{x}_M) \end{bmatrix} = \mathbf{0}, \tag{9}$$

where $\mathbf{U}_i^{\eta,*} := [\mathbf{u}_i^\eta(\mathbf{x}_1), \mathbf{u}_i^\eta(\mathbf{x}_2), \ldots, \mathbf{u}_i^\eta(\mathbf{x}_M)]$ is the solution we seek, $\tilde{\mathbf{G}}_i^\eta := [\mathbf{g}_i^\eta(\mathbf{x}_1)/\mathbf{F}_1[\mathbf{b}^\eta](\mathbf{x}_1), \mathbf{g}_i^\eta(\mathbf{x}_2)/\mathbf{F}_1[\mathbf{b}^\eta](\mathbf{x}_2), \ldots, \mathbf{g}_i^\eta(\mathbf{x}_M)/\mathbf{F}_1[\mathbf{b}^\eta](\mathbf{x}_M)]$ is the reparameterized loading vector, and $\mathbf{G}_i^\eta := [\mathbf{g}_i^\eta(\mathbf{x}_1), \mathbf{g}_i^\eta(\mathbf{x}_2), \ldots, \mathbf{g}_i^\eta(\mathbf{x}_M)]$ is the original loading vector. Here, we notice that all task-specific information are encoded in $\tilde{\mathbf{G}}_i^\eta$ and can be captured in the lifting layer parameter. Therefore, when seeing equation 9 as an implicit problem of $\mathbf{U}_i^{\eta,*}$ and $\tilde{\mathbf{G}}_i^\eta$, it is actually independent of the task parameter field $\mathbf{b}^\eta$, i.e., this problem is task-independent. In the later contents we refer to equation 9 without the task index, as $\mathcal{H}(\mathbf{U}^*; \tilde{\mathbf{G}})$ for notation simplicity.

To solve for $\mathbf{U}^*$ from the nonlinear system $\mathcal{H}(\mathbf{U}^*; \tilde{\mathbf{G}}) = \mathbf{0}$, a popular approach would be to use fixed-point iteration methods such as the Newton-Raphson method. In particular, with an initial guess of the solution (denoted as $\mathbf{U}^0$), the process is repeated to produce successively better approximations to the roots of equation 9, from the solution of iteration $l$ (denoted as $\mathbf{U}^l$) to the solution of iteration $l+1$ (denoted as $\mathbf{U}^{l+1}$) as:

$$\mathbf{U}^{l+1} = \mathbf{U}^l - (\nabla\mathcal{H}(\mathbf{U}^l; \tilde{\mathbf{G}}))^{-1}\mathcal{H}(\mathbf{U}^l; \tilde{\mathbf{G}}) := \mathbf{U}^l + \mathcal{R}(\mathbf{U}^l, \tilde{\mathbf{G}}), \tag{10}$$

until a sufficiently precise value is reached. In the following, we show that as long as Assumptions 1 and 2 hold, i.e., there exists a converging fixed point method, then MetaP can be seen as an resemblance of the fixed point method in equation 10 and hence acts as an universal approximator of the solution operator for equation 8. Assumptions 1 and 2 ensure the hidden PDEs to be numerically solvable with a converging iterative solver, which is a required condition of most numerical PDE solving problems. Then, taking $\mathbf{U}^0 := [\mathbf{x}_1, \cdots, \mathbf{x}_M]$ as the initial guess, we aim to show that for any desired accuracy $\varepsilon > 0$, one can find a sufficiently large $L > 0$ and sets of parameters $\theta^\eta = \{\theta_P^\eta, \theta_I, \theta_Q\}$, such that the resultant MetaP model acts as a fixed point method and its prediction satisfies $\left\| \mathcal{Q}_{\theta_Q} \circ (\mathcal{J}_{\theta_I})^L \circ \mathcal{P}_{\theta_P^\eta}([\mathbf{U}^0, \mathbf{G}^\eta]^\mathrm{T}) - \mathbf{U}^{\eta,*} \right\|_{l^2(\mathbb{R}^M)} \le \varepsilon$ for all tasks and samples.

**Assumption 1.** *There exists a fixed point equation, $\mathbf{U} = \mathbf{U} + \mathcal{R}(\mathbf{U}, \tilde{\mathbf{G}})$ for the implicit problem equation 9, such that $\mathcal{R} : \mathbb{R}^{2M} \mapsto \mathbb{R}^M$ is a continuous function satisfying $\mathcal{R}(\mathbf{U}, \tilde{\mathbf{G}}) = \mathbf{0}$ and $\|\mathcal{R}(\hat{\mathbf{U}}, \tilde{\mathbf{G}}) - \mathcal{R}(\tilde{\mathbf{U}}, \tilde{\mathbf{G}})\|_{l^2(\mathbb{R}^M)} \le m\|\hat{\mathbf{U}} - \tilde{\mathbf{U}}\|_{l^2(\mathbb{R}^M)}$ for any two vectors $\hat{\mathbf{U}}, \tilde{\mathbf{U}} \in \mathbb{R}^M$. Here, $m > 0$ is a constant independent of $\tilde{\mathbf{G}}$.*

**Assumption 2.** *With the initial guess $\mathbf{U}^0 := [\mathbf{x}_1, \cdots, \mathbf{x}_M]$, the fixed-point iteration $\mathbf{U}^{l+1} = \mathbf{U}^l + \mathcal{R}(\mathbf{U}^l, \tilde{\mathbf{G}})$ $(l = 0, 1, \dots)$ converges, i.e., for any given $\varepsilon > 0$, there exists an integer $L$ such that*

$$\|\mathbf{U}^l - \mathbf{U}^*\|_{l^2(\mathbb{R}^M)} \le \varepsilon, \quad \forall l > L,$$

*for all possible input instances $\tilde{\mathbf{G}} \in \mathbb{R}^M$ and their corresponding solutions $\mathbf{U}^*$.*

Then, we have our universal approximation theorem as below, with proof provided in Appendix A:

**Theorem 1** (Universal approximation). *Given Assumptions 1-2, let the activation function $\sigma$ for all iterative kernel integration layers be the ReLU function, and the activation function in the projection layer be the identity function. Then for any $\varepsilon > 0$, there exist sufficiently large layer number $L > 0$ and feature dimension number $d_h > 0$, such that one can find a parameter set for the multi-task problem, $\theta^\eta = [\theta_P^\eta, \theta_I, \theta_Q]$, such that the corresponding MetaP model satisfies*

$$\left\| \mathcal{Q}_{\theta_Q} \circ (\mathcal{J}_{\theta_I})^L \circ \mathcal{P}_{\theta_P^\eta}([\mathbf{U}^0, \mathbf{G}^\eta]^\mathrm{T}) - \mathbf{U}^{\eta,*} \right\| \le \varepsilon, \quad \forall \mathbf{G}^\eta \in \mathbb{R}^M,$$

*for all tasks.*

## 4 EXPERIMENTS

In this section, we demonstrate the empirical effectiveness of the proposed MetaP approach. Specifically, we conduct experiments on a synthetic dataset from a nonlinear PDE solving problem, a benchmark dataset of heterogeneous materials subject to large deformation, and a real-world dataset from biological tissue mechanical testing, and compare the proposed method against competitive GBML baselines. All of the experiments are tested using PyTorch with Adam optimizer, with detailed settings provided in the Appendix B. In all tests we considered the averaged relative error, $\|\mathbf{u}_{i,pred} - \mathbf{u}_i\|_{L^2(\Omega)} / \|\mathbf{u}_i\|_{L^2(\Omega)}$, as the error metric (lower means better). We have repeated each experiment for 5 times, and reported the averaged errors and their standard errors.

### 4.1 SYNTHETIC DATA SETS AND ABLATION STUDY

We first consider the PDE solution finding problem of the Holzapfel-Gasser-Odgen (HGO) model (Holzapfel et al., 2000), which describes the deformation of hyperelastic, anisotropic, and fiber-reinforced materials. Different tasks correspond to different material parameter sets, $\{k_1, k_2, E, \nu, \alpha\}$, where $k_1$ and $k_2$ are fiber modulus and the exponential coefficient, respectively, $E$ is the Young's modulus, $\nu$ is the Poisson ratio, and $\alpha$ is the fiber angle direction from the reference direction. In this example the physical response of interests is the displacement field $\mathbf{u} : [0,1]^2 \to \mathbb{R}^2$ subject to different traction loadings applied on the top edge of this material. Therefore, we take the input function $\mathbf{g}(\mathbf{x})$ as the padded traction loading field, and the output function as the corresponding displacement field.

To investigate the performance of MetaP in few-shot learning, we generate 60 tasks for training, validation, and 1 in-distribution (ID) test task by sampling the physical parameters $k_1, k_2 \sim \mathcal{U}[0.1, 1]$, $E \sim \mathcal{U}[0.5, 1.5]$, $\nu \sim \mathcal{U}[0.1, 0.49]$, and $\alpha \sim \mathcal{U}[\pi/10, \pi/2]$. Here $\mathcal{U}$ stands for uniform distribution. To further evaluate the generalizability when the physical parameters in test tasks are outside the

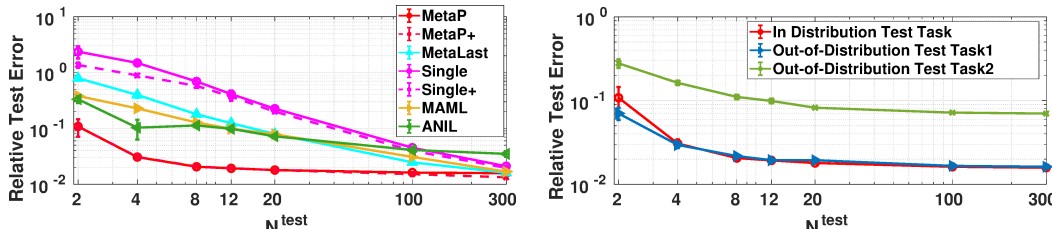

Figure 2: Results on a synthetic data set. Left: The ablation study comparison on test errors in the in-distribution test. Right: The relative error of MetaP in in-distribution and out-of distribution tests.

training regime, we also generate 2 out-of-distribution (OOD) test tasks with physical parameters from different distributions. For the first OOD task we sample $E \sim \mathcal{U}[1.5, 2]$, while for the second OOD task $E \sim \mathcal{U}[0.3, 0.5]$. As show in Figure 5 of Appendix B, the first OOD task (denoted as "OOD Task1") corresponds to a stiffer material sample and smaller deformation for each given loading, while the second OOD task (denoted as "OOD Task2") generates a softer material sample and larger deformation. For each training task, we generate 500 data pairs $\mathcal{D}^{\eta} := \{(\mathbf{g}_i^{\eta}, \mathbf{u}_i^{\eta})\}_{i=1}^{500}$, by sampling the vertical traction loading from a Gaussian random field. Then, the corresponding ground-truth displacement field is obtained using the finite element method implemented in FEniCS (Alnæs et al., 2015). For the test tasks, we train with $N^{\text{test}} = \{2, 4, 8, 12, 20, 100, 300\}$ numbers of labelled data pairs (the context set), and evaluate the resultant model on an additional dataset with 200 data pairs (the target set). An 8-layer IFNO is employed as the base model.

**Ablation Study.** We first conduct an ablation study with three settings. 1) Follow the meta-train and meta-test phases as in Algorithm 1, with task-wise adaptation only to the lifting layer in both phases (denotes as "MetaP"). 2) After MetaP, perform an additional fine-tuning step to all parameters in the meta-test phase (denotes as "MetaP+"). With this test, we aim to investigate if our algorithm has successfully identified all the common features in the iterative and projection layers. 3) Apply task-wise adaptation only to the projection layer in both meta-train and meta-test phases (denoted as "MetaLast"). With this test, we study if the successful "adapting last layers" strategy in image classification problems would also apply for our PDE solving problem. Besides these three settings, we also report the few-shot learning results with four baseline methods: 1) Learn a neural operator model only based on the context data set on the test task (denoted as "Single"), 2) Pretrain a single neural operator model based on all training task data sets, then fine-tune it based on the context test task data set (denoted as "Single+"), 3) MAML, and 4) ANIL. As shown in the left plot of Figure 2, MetaP and MetaP+ are both able to quickly adapt with few data pairs – to achieve a test error below 5%, "Single" and "Single+" require 100 data pairs, while MetaP and MetaP+ requires only 4 data pairs. On the other hand, MetaLast, MAML and ANIL have similar performance. They all require 100 data pairs to achieve a $< 5\%$ test error. This observation verifies our theoretical analysis: on the multi-task parametric PDE solution operator learning problem, one should adapt the first layer, not the last ones. Moreover, when comparing MetaP and MetaP+ we can see that the additional fine-tune step barely improves the performance, especially in the few-sample regime. This fact verifies the efficacy of MetaP, and indicates that our method has successfully capture the underlying task diversity by adapting the first layer, so no further fine-tuning is required.

**In-Distribution and Out-Of-Distribution Tests.** On the right plot of Figure 2, we demonstrate the relative test error of MetaP in both ID and OOD tasks. We can see that these three test errors are both in a similar scale as the error on training tasks. The error from OOD task1 is slightly smaller than the ID test task error, while the error from OOD task2 is much larger, probably due to the fact that the solutions in OOD task1 generally have smaller magnitude and hence its solution operator lies more in a linear regime, which makes the solution operator learning task easier. These results validates the good generalization performance of MetaP.

## 4.2 BENCHMARK MECHNICAL MNIST DATASETS

To further test the capability of MetaP on benchmark datasets, we test MetaP and four baseline methods on Mechanical MNIST (Lejeune, 2020). Mechanical MNIST is a dataset of heterogeneous material undergoing large deformation. It contains 70,000 heterogeneous material specimens, and each specimen is governed by the Neo-Hookean material with a varying modulus converted from

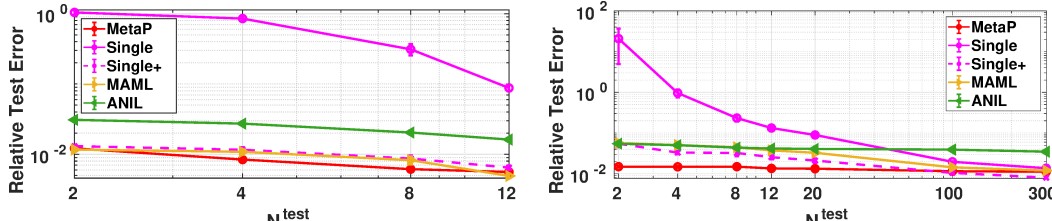

Figure 3: Comparison of MetaP and four baseline methods on the benchmark dataset (MechnicalM-NIST, left plot) and the real-world dataset (heart valve tissue, right plot).

the MNIST bitmap images. On each specimen, 32 loading/response data pairs are provided[3]. Here in, we randomly select 101 specimens for training and validation, and a new and unseen specimen as the test task. On the meta-test phase, we reserve 20 data pairs on the test task as the target set for evaluation, then train each model under the few-shot learning setting with $N^{\text{test}} = \{2, 4, 8, 12\}$ labelled data pairs as the context set. All approaches are developed based on an 8-layer IFNO model.

We present the results in the left plot of Figure 3. The neural operator model learned by MetaP again outperforms the state-of-the-art GBML models. Our MetaP model achieves $1\%$ when using only 2 labelled data pair on the test task, while the "Single" model has around $100\%$ error even using 8 labelled data pairs, due to overfitting. This fact highlights the importance of learning across multi-tasks in engineering applications – when the total number of measurements on each specimen is limited, it is necessary to transfer the knowledge across specimens. Moreover, we notice that in this example ANIL is the least effective GBML method, which is even less efficient than the pretrained model ("Single+"), probably due to the inefficacy of the adapting last layers strategy.

## 4.3 APPLICATION ON REAL-WORLD DATA SETS

We now take a step further to demonstrate the performance of our method on a real-world physical response dataset which is NOT generated by sovling PDEs. We consider the problem of learning the mechanical response of multiple biological tissue specimens from DIC displacement tracking measurements. As demonstrated in Figure 1, we measure the biaxial loading of tricuspid valve anterior leaflet (TVAL) specimens from a porcine heart, such that each specimen (as a task) corresponds to a different region of the leaflet. Due to the material heterogeneity of biological tissues, these specimens are with different mechanical and structural properties.

In this task, we aim to model the tissue response by learning a neural operator mapping the boundary displacement loading to the interior displacement field on each tissue specimen. On each specimen, we have 500 available data pairs. Due to the challenges of obtaining the experimental tissue, only 14 specimens are available in total. This example also stands for a common challenging setting in real-world applications: we not only have the few-shot learning challenge, but also suffer from the difficulty from limited available training tasks. With a 4-layer IFNO as the base model, we train each model based on $N^{\text{test}} \in [2, 300]$ samples, then evaluate the performance on another 200 samples. The results are provided on the right plot of Figure 3. MetaP performs the best with low data samples among all the methods, and still beat our MAML and ANIL variants when $N^{\text{test}} = 300$. Interestingly, MAML and ANIL did not even beat the "Single+" method, possibly due to the low efficacy of the adapting last layers strategy and the small number of training tasks.

## 5 CONCLUSION

In this paper we propose MetaP, the first neural-operator-based meta-learning approach that are designed to achieve good transferability in learning complex physical system responses with significant improvement in sample efficiency. The first layer adaption used by our method is theoretically motivated and shown to be the universal solution operator for multiple parametric PDE solving tasks. We demonstrate the effectiveness of our proposed MetaP algorithm on various synthetic and real-world datasets, showing promises over baseline methods. For future work, we will investigate the applicability of the proposed approach to other neural operators.

---

[3]We have excluded small deformation samples with the maximum displacement magnitude $\leq 0.1$.

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

## A    PROOF OF THEOREM 1

In this section we provide the detailed proof for Theorem 1, based on Assumptions 1 and 2. Intuitively, that means the underlying implicit problem is solved with a converging fixed point method. This condition is a basic requirement by numerical PDEs, and it generally holds true in many applications governed by nonlinear and complex PDEs, such as in our three experiments.

Here, we prove that the MetaP is universal, i.e., give a fixed point method satisfying Assumptions 1-2, one can find parameter sets $\theta^\eta$ whose output approximates $\mathbf{U}^{\eta,*}$ to a desired accuracy, $\varepsilon > 0$, for all $\eta = 1, \cdots, H$ tasks. For the task-wise parameters, with a slight abuse of notation, we denote $P^\eta \in \mathbb{R}^{d_h M \times (d_g + s) M}$ as the collection of the pointwise weight matrices at each discretization point in $\chi$ for the $\eta$-th task, and $\mathbf{p}^\eta \in \mathbb{R}^{d_h M}$ for the bias in the lifting layer. Then, for the parameters shared among all tasks, in the iterative layer we denote $\mathbf{C} = [\mathbf{c}(\mathbf{x}_1), \cdots, \mathbf{c}(\mathbf{x}_M)] \in \mathbb{R}^{d_h M}$ as the collection of pointwise bias vectors $\mathbf{c}(\mathbf{x}_i)$, $W \in \mathbb{R}^{d_h \times d_h}$ for the local linear transformation, and $R = \mathcal{F}[\kappa(\cdot; \mathbf{v})] \in \mathbb{C}^{d_h \times d_h \times M} \in \mathbb{C}^{d_h \times d_h \times M}$ for the Fourier coefficients of the kernel $\kappa$. For simplicity, here we have assumed that the Fourier coefficient is not truncated, and all available frequencies are used. Then, for the projection layer we seek $Q_1 \in \mathbb{R}^{d_Q M \times d_h M}$, $Q_2 \in \mathbb{R}^{d_u M \times d_Q M}$, $\mathbf{q}_1 \in \mathbb{R}^{d_Q M}$ and $\mathbf{q}_2 \in \mathbb{R}^{d_u M}$. For the simplicity of notation, in this section we organize the feature vector $\mathbf{H} \in \mathbb{R}^{d_h M}$ in a way such that the components corresponding to each discretization point are adjacent, i.e., $\mathbf{H} = [\mathbf{H}(\mathbf{x}_1), \cdots, \mathbf{H}(\mathbf{x}_M)]$ and $\mathbf{H}(\mathbf{x}_i) \in \mathbb{R}^{d_h}$.

We point out that under this circumstance, we have the (discretized) iterative layer can be written as

$$\mathcal{J}[\mathbf{H}(l\Delta t)] = \mathbf{H}(l\Delta t) + \Delta t \sigma \left( \tilde{W} \mathbf{H}(l\Delta t) + \mathrm{Re}(\mathcal{F}_{\Delta x}^{-1}(R \cdot \mathcal{F}_{\Delta x}(\mathbf{H}(l\Delta t)))) + \mathbf{C} \right)$$
$$= \mathbf{H}(l\Delta t) + \Delta t \sigma \left( V \mathbf{H}(l\Delta t) + \mathbf{C} \right),$$

with

$$V := \mathrm{Re} \begin{bmatrix} \sum_{n=0}^{M-1} R_{n+1} + W & \sum_{n=0}^{M-1} R_{n+1} \exp\left(\frac{2i\pi\Delta x n}{M}\right) & \cdots & \sum_{n=0}^{M-1} R_{n+1} \exp\left(\frac{2i\pi(M-1)\Delta x n}{M}\right) \\ \sum_{n=0}^{M-1} R_{n+1} \exp\left(\frac{2i\pi\Delta x n}{M}\right) & \sum_{n=0}^{M-1} R_{n+1} + W & \cdots & \sum_{n=0}^{M-1} R_{n+1} \exp\left(\frac{2i\pi(M-2)\Delta x n}{M}\right) \\ \vdots & \vdots & \ddots & \vdots \\ \sum_{n=0}^{M-1} R_{n+1} \exp\left(\frac{2i\pi(M-1)\Delta x n}{M}\right) & \sum_{n=0}^{M-1} R_{n+1} \exp\left(\frac{2i\pi(M-2)\Delta x n}{M}\right) & \cdots & \sum_{n=0}^{M-1} R_{n+1} + W \end{bmatrix}.$$

Here, $R \in \mathbb{C}^{M \times d_h \times d_h}$ with $R_i \in \mathbb{C}^{d_h \times d_h}$ being the component associated with each discretization point $\mathbf{x}_i \in \chi$, $V \in \mathbb{R}^{d_h M \times d_h M}$, $\mathbf{C} \in \mathbb{R}^{d_h M}$, $\tilde{W} := W \oplus W \oplus \cdots \oplus W$ is a $d_h M \times d_h M$ block diagonal matrix formed by $W \in \mathbb{R}^{d_h \times d_h}$, $\mathcal{F}_{\Delta x}$ and $\mathcal{F}_{\Delta x}^{-1}$ denote the discrete Fourier transform and its inverse, respectively. By further taking $R_2 = \cdots = R_M = W = 0$, a $d_h \times d_h$ matrix with all its elements being zero, it suffices to show the universal approximation property for an iterative layer as follows:

$$\mathcal{J}(\mathbf{H}(l\Delta t)) := \mathbf{H}(l\Delta t) + \Delta t \sigma \left( \tilde{V} \mathbf{H}(l\Delta t) + \mathbf{C} \right)$$

where $\tilde{V} := \mathbf{1}_{[M,M]} \otimes V$ with $V \in \mathbb{R}^{d_h \times d_h}$ and $\mathbf{1}_{[m,n]}$ being an $m$ by $n$ all-ones matrix.

To be more precise, we will prove the following theorem:

**Theorem 1** (Universal approximation). *Let* $\mathbf{U}^{\eta,*} = [\mathbf{u}^\eta(\mathbf{x}_1), \mathbf{u}^\eta(\mathbf{x}_2), \ldots, \mathbf{u}^\eta(\mathbf{x}_M)]$ *be the ground-truth solution of $\eta$-th task that satisfies Assumptions 1-2, the activation function $\sigma$ for all iterative kernel integration layers be the ReLU function, and the activation function in the projection layer be the identity function. Then for any $\varepsilon > 0$, there exist sufficiently large layer number $L > 0$ and feature dimension number $d_h > 0$, such that one can find a parameter set for the multi-task problem, $\theta^\eta = [\theta_P^\eta, \theta_I, \theta_Q]$ with the corresponding MetaP model satisfies*

$$\left\| \mathcal{Q}_{\theta_Q} \circ (\mathcal{J}_{\theta_I})^L \circ \mathcal{P}_{\theta_P^\eta}([\mathbf{U}^0, \mathbf{G}^\eta]^{\mathrm{T}}) - \mathbf{U}^{\eta,*} \right\| \leq \varepsilon, \quad \forall \mathbf{G}^\eta \in \mathbb{R}^M.$$

For the proof of this main theorem, we need the following approximation property of a shallow neural network, with its detailed proof provided in You et al. (2022c):

**Lemma 1.** *Given a continuous function $\mathcal{T} : \mathbb{R}^{2M} \mapsto \mathbb{R}^M$, and a non-polynomial and continuous activation function $\sigma$, for any constant $\hat{\varepsilon} > 0$ there exists a shallow neural network model $\hat{\mathcal{T}} := S\sigma(B\mathbf{X} + A)$ such that*

$$||\mathcal{T}(\mathbf{X}) - \hat{\mathcal{T}}(\mathbf{X})||_{l^2(\mathbb{R}^M)} \leq \hat{\varepsilon}, \quad \forall \mathbf{X} \in \mathbb{R}^{2M},$$

*for sufficiently large feature dimension $\hat{d} > 0$. Here, $S \in \mathbb{R}^{M \times \hat{d}M}$, $B \in \mathbb{R}^{\hat{d}M \times 2M}$, and $A \in \mathbb{R}^{\hat{d}M}$ are matrices/vectors which are independent of $\mathbf{X}$.*

We now proceed to the proof of Theorem 1:

*Proof.* Since all $\mathbf{U}^{\eta,*}$ satisfies Assumptions 1-2, for any $\varepsilon > 0$, we first pick a sufficiently large integer $L$ such that the $L$-th layer iteration result of this fixed point formulation satisfies $||\mathbf{U}^L - \mathbf{U}^{\eta,*}||_{l^2(\mathbb{R}^M)} \leq \frac{\varepsilon}{2}$ for all tasks. By taking $\hat{\varepsilon} := \frac{m\varepsilon}{2(1+m)^L}$ in Lemma 1, there exists a sufficiently large feature dimension $\hat{d}$ and one can find $S \in \mathbb{R}^{M \times \hat{d}M}$, $B \in \mathbb{R}^{\hat{d}M \times 2M}$, and $A \in \mathbb{R}^{\hat{d}M}$, such that $\hat{\mathcal{R}}(\mathbf{U}^\eta, \tilde{\mathbf{G}}^\eta) := S\sigma(B[\mathbf{U}^\eta, \tilde{\mathbf{G}}^\eta]^{\mathrm{T}} + A)$ satisfies

$$||\mathcal{R}(\mathbf{U}^\eta, \tilde{\mathbf{G}}^\eta) - \hat{\mathcal{R}}(\mathbf{U}^\eta, \tilde{\mathbf{G}}^\eta)||_{l^2(\mathbb{R}^M)} = ||\mathcal{R}(\mathbf{U}^\eta, \tilde{\mathbf{G}}^\eta) - S\sigma(B[\mathbf{U}^\eta, \tilde{\mathbf{G}}^\eta]^{\mathrm{T}} + A)||_{l^2(\mathbb{R}^M)} \leq \hat{\varepsilon} = \frac{m\varepsilon}{2(1+m)^L},$$

where $m$ is the contraction parameter of $\mathcal{R}$, as defined in Assumption 1. By this construction, we know that $S$ has independent rows. Denoting $\tilde{d} := \hat{d} + 1 > 0$, there exists the right inverse of $S$, which we denote as $S^+ \in \mathbb{R}^{(\tilde{d}-1)M \times M}$, such that

$$SS^+ = I_M, \quad S^+S := \tilde{I}_{(\tilde{d}-1)M},$$

where $I_M$ is the $M$ by $M$ identity matrix, $\tilde{I}_{(\tilde{d}-1)M}$ is a $(\tilde{d}-1)M$ by $(\tilde{d}-1)M$ block matrix with each of its element being either 1 or 0. Hence, for any vector $Z \in \mathbb{R}(\tilde{d}-1)M$, we have $\sigma(\tilde{I}_{(\tilde{d}-1)M}Z) = \tilde{I}_{(\tilde{d}-1)M}\sigma(Z)$. Moreover, we note that $S$ has a very special structure: from the $((i-1)(\tilde{d}-1)+1)$-th to the $(i(\tilde{d}-1))$-th column of $S$, all nonzero elements are on its $i$-th row. Correspondingly, we can also choose $S^+$ to have a special structure: from the $((i-1)(\tilde{d}-1)+1)$-th to the $(i(\tilde{d}-1))$-th row of $S^+$, all nonzero elements are on its $i$-th column. Hence, when multiplying $S^+$ with $\mathbf{U}$, there will be no entanglement between different components of $\mathbf{U}$. That means, $S^+$ can be seen as a pointwise weight function.

We now construct the MetaP as follows. In this construction, we choose the feature dimension as $d_h := \tilde{d}M$. With the input $[\mathbf{U}^0, \mathbf{G}^\eta] \in \mathbb{R}^{2M}$, for the lift layer we set

$$P^\eta := \mathbf{1}_{[M,1]} \otimes \begin{bmatrix} S^+ & \mathbf{0} \\ \mathbf{0} & D^\eta \end{bmatrix} = \underbrace{\begin{bmatrix} S^+ & \mathbf{0} & S^+ & \mathbf{0} & \cdots & S^+ & \mathbf{0} \\ \mathbf{0} & D^\eta & \mathbf{0} & D^\eta & \cdots & \mathbf{0} & D^\eta \end{bmatrix}^{\mathrm{T}}}_{\text{repeated for } M \text{ times}} \in \mathbb{R}^{d_h M \times 2M},$$

and $\mathbf{p}^\eta := \mathbf{0} \in \mathbb{R}^{d_h M}$. Here, $D^\eta := \mathrm{diag}[1/\mathbf{F}_1[\mathbf{b}^\eta](\mathbf{x}_1), \cdots, 1/\mathbf{F}_1[\mathbf{b}^\eta](\mathbf{x}_M)]$. As such, the initial layer of feature is then given by

$$\mathbf{H}(0) = P^\eta([\mathbf{U}^0, \mathbf{G}^\eta]^{\mathrm{T}}) = \mathbf{1}_{[M,1]} \otimes [S^+\mathbf{U}^0, D^\eta\mathbf{G}^\eta]^{\mathrm{T}} = \mathbf{1}_{[M,1]} \otimes [S^+\mathbf{U}^0, \tilde{\mathbf{G}}^\eta]^{\mathrm{T}} \in \mathbb{R}^{dM}.$$

Here, we point out that $P^\eta$ and $\mathbf{p}^\eta$ can be seen as pointwise weight and bias functions, respectively.

Next we construct the shared iterative layer $\mathcal{J}$, by setting

$$V := \begin{bmatrix} \tilde{I}_{(\tilde{d}-1)M}B/M \\ 0 \end{bmatrix} \begin{bmatrix} S/\Delta t & \mathbf{0} \\ \mathbf{0} & I_M/\Delta t \end{bmatrix}, \quad \tilde{V} := \mathbf{1}_{[M,M]} \otimes V, \text{ and } C := \mathbf{1}_{[M,1]} \otimes \begin{bmatrix} \tilde{I}_{(\tilde{d}-1)M}A/\Delta t \\ \mathbf{0} \end{bmatrix}.$$

Note that $\tilde{V}$ is independent of $\eta$, and falls into the formulation of $V$, by letting $R_1 = V$ and $R_2 = R_2 = \cdots = R_M = W = 0$. For the $l+1$-th layer of feature vector, we then arrive at

$$\mathbf{H}((l+1)\Delta t) = \mathbf{H}(l\Delta t) + \Delta t\sigma\left(\tilde{V}\mathbf{H}(l\Delta t) + \mathbf{C}\right)$$

$$= \mathbf{H}(l\Delta t) + \left(I_M \otimes \begin{bmatrix} S^+S & \mathbf{0} \\ \mathbf{0} & I_M \end{bmatrix}\right) \sigma\left(\left(\mathbf{1}_{[M,1]} \otimes \begin{bmatrix} B/M \\ \mathbf{0} \end{bmatrix}\right)\left(\mathbf{1}_{[1,M]} \otimes \begin{bmatrix} S & \mathbf{0} \\ \mathbf{0} & I_M \end{bmatrix}\right)\mathbf{H}(l\Delta t) + \mathbf{1}_{[M,1]} \otimes \begin{bmatrix} A \\ \mathbf{0} \end{bmatrix}\right),$$

where $\mathbf{H}(l\Delta t) = [\hat{\mathbf{h}}_1^{l\Delta t}, \hat{\mathbf{h}}_2^{l\Delta t}, \ldots, \hat{\mathbf{h}}_{2M-1}^{l\Delta t}, \hat{\mathbf{h}}_{2M}^{l\Delta t}]^{\mathrm{T}}$ denotes the (spatially discretized) hidden layer feature at the $l-$th iterative layer of the IFNO. Subsequently, we note that the second part of the feature vector, $\hat{\mathbf{h}}_{2j}^{l\Delta t} \in \mathbb{R}^M$, satisfies

$$\hat{\mathbf{h}}_{2j}^{(l+1)\Delta t} = \hat{\mathbf{h}}_{2j}^{l\Delta t} = \cdots = \hat{\mathbf{h}}_{2j}^0 = \tilde{\mathbf{G}}^\eta, \quad \forall l = 0, \cdots, L-1, \forall j = 1, \cdots, M$$

Hence, the first part of the feature vector, $\hat{\mathbf{h}}_{2j-1}^{l\Delta t} \in \mathbb{R}^{(\tilde{d}-1)M}$, satisfies the following iterative rule:

$$\hat{\mathbf{h}}_{2j-1}^{(l+1)\Delta t} = \hat{\mathbf{h}}_{2j-1}^{l\Delta t} + S^+ S\sigma(B[S\hat{\mathbf{h}}_{2j-1}^{l\Delta t}, \tilde{\mathbf{G}}^\eta]^{\mathrm{T}} + A), \quad \forall l = 0, \cdots, L-1, \forall j = 1, \cdots, M,$$

and

$$\hat{\mathbf{h}}_1^{(l+1)\Delta t} = \hat{\mathbf{h}}_3^{(l+1)\Delta t} = \cdots = \hat{\mathbf{h}}_{2M-1}^{(l+1)\Delta t}.$$

Finally, for the projection layer $\mathcal{Q}$, we set the activation function in the projection layer as the identity function, $Q_1 := I_{d_h M}$ (the identity matrix of size $d_h M$), $Q_2 := [S, \mathbf{0}] \in \mathbb{R}^{M \times d_h M}$, $\mathbf{q}_1 := \mathbf{0} \in \mathbb{R}^{d_h M}$, and $\mathbf{q}_2 := \mathbf{0} \in \mathbb{R}^M$. Denoting the output $\mathbf{U}^\eta := \mathcal{Q}_{\theta_Q} \circ (\mathcal{J}_{\theta_I})^L \circ \mathcal{P}_{\theta_P}^\eta([\mathbf{U}^0, \mathbf{G}^\eta]^{\mathrm{T}})$, we now show that $\mathbf{U}^\eta$ can approximate $\mathbf{U}^{\eta,*}$ with a desired accuracy $\varepsilon$:

$$\begin{aligned}
\|\mathbf{U}^\eta - \mathbf{U}^{\eta,*}\| &\leq \|\mathbf{U}^\eta - \mathbf{U}^L\|_{l^2(\mathbb{R}^M)} + \|\mathbf{U}^L - \mathbf{U}^{\eta,*}\|_{l^2(\mathbb{R}^M)} \\
&\leq \|S\hat{\mathbf{h}}_1^{L\Delta t} - \mathbf{U}^L\|_{l^2(\mathbb{R}^M)} + \frac{\varepsilon}{2} \quad \text{(by Assumption 2)} \\
&\leq \|S\hat{\mathbf{h}}_1^{(L-1)\Delta t} - \mathbf{U}^{L-1}\|_{l^2(\mathbb{R}^M)} + \|\hat{\mathcal{R}}(S\hat{\mathbf{h}}_1^{(L-1)\Delta t}, \tilde{\mathbf{G}}) - \mathcal{R}(\mathbf{U}^{L-1}, \tilde{\mathbf{G}})\|_{l^2(\mathbb{R}^M)} + \frac{\varepsilon}{2} \\
&\leq \|S\hat{\mathbf{h}}_1^{(L-1)\Delta t} - \mathbf{U}^{L-1}\|_{l^2(\mathbb{R}^M)} + \|\hat{\mathcal{R}}(S\hat{\mathbf{h}}_1^{(L-1)\Delta t}, \tilde{G}b) - \mathcal{R}(S\hat{\mathbf{h}}_1^{(L-1)\Delta t}, \tilde{G}b)\|_{l^2(\mathbb{R}^M)} \\
&\quad + \|\mathcal{R}(S\hat{\mathbf{h}}_1^{(L-1)\Delta t}, \tilde{G}b) - \mathcal{R}(\mathbf{U}^{L-1}, \tilde{G}b)\|_{l^2(\mathbb{R}^M)} + \frac{\varepsilon}{2} \\
&\leq (1+m)\|S\hat{\mathbf{h}}_1^{(L-1)\Delta t} - \mathbf{U}^{L-1}\|_{l^2(\mathbb{R}^M)} + \frac{m\varepsilon}{2(1+m)^L} + \frac{\varepsilon}{2} \quad \text{(by Lemma 1 and Assumption 1)} \\
&\leq \frac{m\varepsilon}{2(1+m)^L}(1 + (1+m) + (1+m)^2 + \cdots + (1+m)^{L-1}) + \frac{\varepsilon}{2} \\
&\leq \frac{\varepsilon}{2} + \frac{\varepsilon}{2} = \varepsilon.
\end{aligned}$$

$\square$

# B  DATA GENERATION AND TRAINING DETAILS

In the following we briefly describe the empirical process of generating datasets, and the settings employed in running of each algorithm. For a fair comparison, for each algorithm, we tune the hyperparameters, including the learning rate from $\{0.1, 0.01, 0.001, 0.0001, 0.00001, 0.000001\}$, the decay rate from $\{0.5, 0.7, 0.9\}$, the weight decay parameter from $\{0.01, 0.001, 0.0001, 0.00001, 0.000001\}$, and the inner loop learning rate for MAML and ANIL from $\{0.01, 0.001, 0.0001, 0.00001, 0.000001\}$, to minimize the error on a separate validation dataset. In all experiments we decrease the learning rate with a ratio of learning rate decay rate every 100 epochs. The code and the processed datasets will be publicly released at github for readers to reproduce the experimental results.

## B.1  EXAMPLE 1: SYNTHETIC DATA SETS

### B.1.1  DATA GENERATION

In the synthetic data example, we consider the modeling problem of a hyperelastic, anisotropic, fiber-reinforced material, and seek to find its displacement field $\mathbf{u} : [0, 1]^2 \to \mathbb{R}^2$ under different boundary loadings. In this problem, the specimen is assumed to be subject to a uniaxial tension $T_y(\mathbf{x})$ on the top edge (see Figure 4(a)). To generate training and test samples, the Holzapfel-Gasser-Odgen (HGO) model (Holzapfel et al., 2000) was employed to describe the constitutive behavior of the material in this example, with its strain energy density function given as:

$$\begin{aligned}
\eta =\ & \frac{E}{4(1+\nu)}(\bar{I}_1 - 2) - \frac{E}{2(1+\nu)}\ln(J) \\
& + \frac{k_1}{2k_2}\left(\exp\left(k_2\langle S(\alpha)\rangle^2\right) + \exp\left(k_2\langle S(-\alpha)\rangle^2\right) - 2\right) + \frac{E}{6(1-2\nu)}\left(\frac{J^2-1}{2} - \ln J\right).
\end{aligned}$$

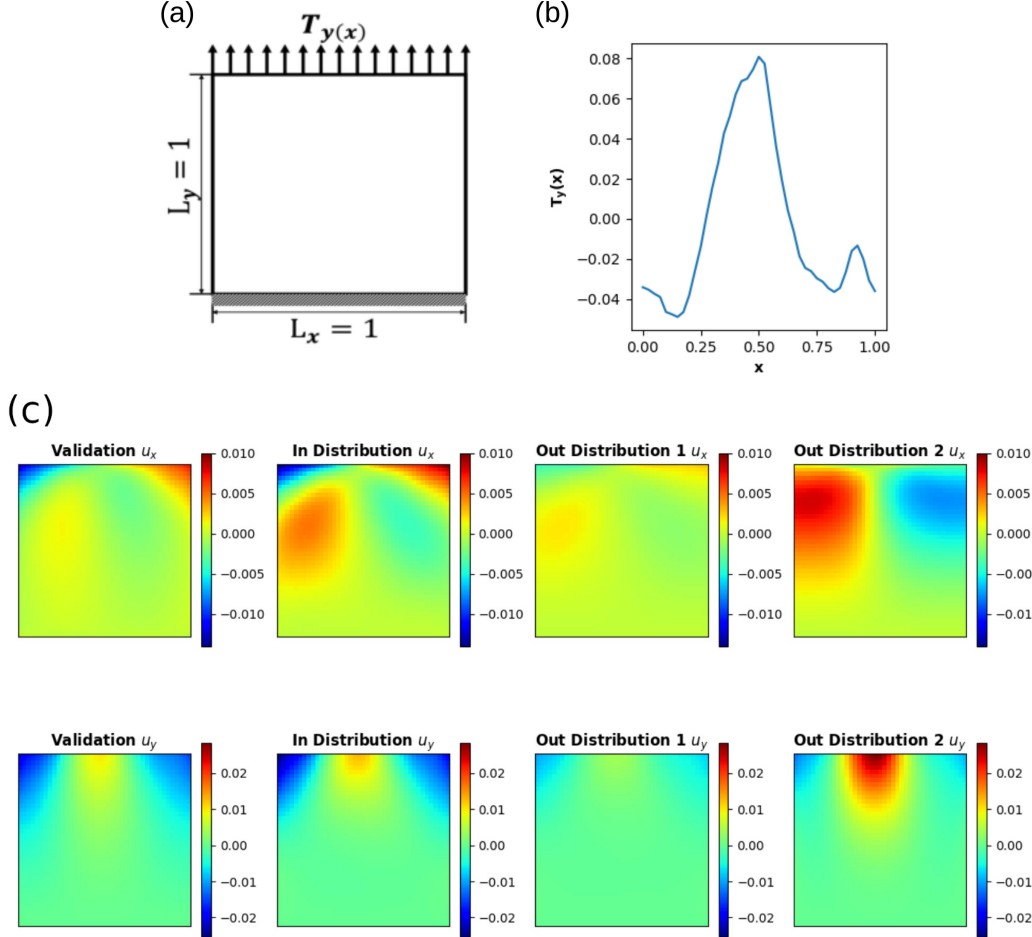

Figure 4: Problem setup of example 1: the synthetic data sets. (a) A unit square specimen subject to uniaxial tension with Neumann-type boundary condition. (b) & (c) Visualization of an instances of the loading field $T_y(x)$, and the corresponding ground-truth solutions $\mathbf{u}^\eta(\mathbf{x})$ from the in-distribution and out-of-distribution tasks, showing the solution diversity across different tasks, due to the change of underlying hidden material parameter set.

Here, $\langle \cdot \rangle$ denotes the Macaulay bracket, and the fiber strain of the two fiber groups is defined as:

$$S(\alpha) = \frac{\overline{I}_4(\alpha) - 1 + |\overline{I}_4(\alpha) - 1|}{2}.$$

where $k_1$ and $k_2$ are fiber modulus and the exponential coefficient, respectively, $c_{10}$ is the moduli for the non-fibrous ground matrix, $E$ is the Young's modulus, and $\nu$ is the Poisson ratio. Moreover, $\overline{I}_1 = \text{tr}(\mathbf{C})$ is the is the first invariant of the right Cauchy-Green tensor $\mathbf{C} = \mathbf{F}^T\mathbf{F}$, $\mathbf{F}$ is the deformation gradient, and $J$ is related with $\mathbf{F}$ such that $J = \det \mathbf{F}$. For the fiber group with angle direction $\alpha$ from the reference direction, $\overline{I}_4(\alpha) = \mathbf{n}^T(\alpha)\mathbf{C}\mathbf{n}(\alpha)$ is the fourth invariant of the right Cauchy-Green tensor $\mathbf{C}$, where $\mathbf{n}(\alpha) = [\cos(\alpha), \sin(\alpha)]^T$. To generate samples for different specimens,different specimens (tasks) correspond to different material parameter sets, $\{k_1, k_2, E, \nu, \alpha\}$. For the training tasks, the validation task, and the in-distribution (ID) test task, their physical parameters are sampled from: $k_1, k_2 \sim \mathcal{U}[0.1, 1]$, $E \sim \mathcal{U}[0.55, 1.5]$, $\nu \sim \mathcal{U}[0.01, 0.49]$, and $\alpha \sim \mathcal{U}[\pi/10, \pi/2]$. For the two out-of-distribution (OOD) test tasks, we sample their parameters following $k_1, k_2 \sim \mathcal{U}[1, 1.9]$, $E \sim \mathcal{U}[1.5, 2] \cup \mathcal{U}[0.5, 0.55]$, $\nu \sim \mathcal{U}[0.01, 0.49]^4$, and $\alpha \sim \mathcal{U}[\pi/2, 3\pi/4] \cup [0, \pi/10]$. To generate the high-fidelity (ground-truth) dataset, we sampled $500$ different vertical traction conditions $T_y(\mathbf{x})$ on the top edge from a random field, following the algorithm in Lang & Potthoff (2011); Yin et al. (2022b). In particular, $T_y(\mathbf{x})$ is taken as the restriction of a 2D random field, $\phi(\mathbf{x}) = \mathcal{F}^{-1}(\gamma^{1/2}\mathcal{F}(\Gamma))(\mathbf{x})$, on the top edge. Here, $\Gamma(\mathbf{x})$ is a Gaussian white noise random field on $\mathbb{R}^2$, $\gamma = (w_1^2 + w_2^2)^{-\frac{5}{4}}$ represents a correlation function, and $w_1, w_2$ are the wave numbers on $x$ and $y$ directions, respectively. Then, for each sampled traction loading, we solved the displacement field on the entire domain by minimizing potential energy using the finite element method implemented in FEniCS (Alnæs et al., 2015). In particular, the displacement filed was approximated by continuous piecewise linear finite elements with triangular mesh, and the grid size was taken as $0.025$. Then, the finite element solution was interpolated onto $\chi$, a structured $41 \times 41$ grid which will be employed as the discretization in our neural operators.

To visualize the domain characteristics for tasks, the distribution of each parameter for training, validation and test tasks are demonstrated in Figure 5, and the corresponding solution fields are plotted in Figure 4(c), showing the diversity across different tasks due to the change of underlying hidden material parameter set, $\{k_1, k_2, E, \nu, \alpha\}$. From Figures 5 and 4(c), one can see that OOD Task1 corresponds a stiffer material (with large Young's modulus $E$) and hence smaller deformation subject to the same loading $T_y(\mathbf{x})$. On the other hand, OOD Task2 corresponds a softer material (with small Young's modulus $E$) and larger deformation. Therefore, the material response of OOD Task1 specimen is more likely to lie in a linear region, which is easier to learn and explains the relatively small test error on this task. On the other hand, the material response of OOD Task2 is more nonlinear and hence complex due to larger deformation, as shown in Figure 4(c), and results in the relatively larger test error in Figure 2.

### B.1.2 ALGORITHM SETTINGS

**Base model**: As the base model for all algorithms, we construct an architecture for IFNO (You et al., 2022c) as follows. First, the input loading field instance $\mathbf{g}(\mathbf{x}) \in \mathcal{A}$ is lifted to a higher dimensional representation via lift layer $\mathcal{P}[\mathbf{g}](\mathbf{x})$, which is parameterized as a 1-layer feed forward linear layer with width (3,32). Then for the iterative layer in equation 1, we implement $\mathcal{F}^{-1}[\mathcal{F}[\kappa(\cdot; \mathbf{v})] \cdot \mathcal{F}[\mathbf{h}(\cdot, l\Delta t)]](\mathbf{x})$ with 2D fast Fourier transform (FFT) with input channel and output channel widths both set as 32 and the truncated Fourier modes set as 8. The local linear transformation parameter, $W$, is parameterized as a 1-layer feed forward network with width (32,32). In the projection layer, a 2-layer feed forward network with width (32,128,2) is employed. To accelerate the training procedure, we apply the shallow-to-deep training technique to initialize the optimization problem. In particular, we start from the NN model with depth $L = 1$, train until the loss function reaches a plateau, then use the resultant parameters to initialize the parameters for the next depth, with $L = 2$, $L = 4$, and $L = 8$. In the synthetic experiments, we set the layer depth as $L = 8$.

**MetaP**: We split the total 60 training tasks to two groups: 59 tasks for the purpose of training and 1 task for the purpose of validation. During the meta-train phase, we train for the task-wise parameters

---

[4]Here we sample both ID and OOD tasks from the same range of $\nu$, due to the fact that $[0.01, 0.49]$ is the range of Poisson ratio for common materials (Bischofs & Schwarz, 2005).

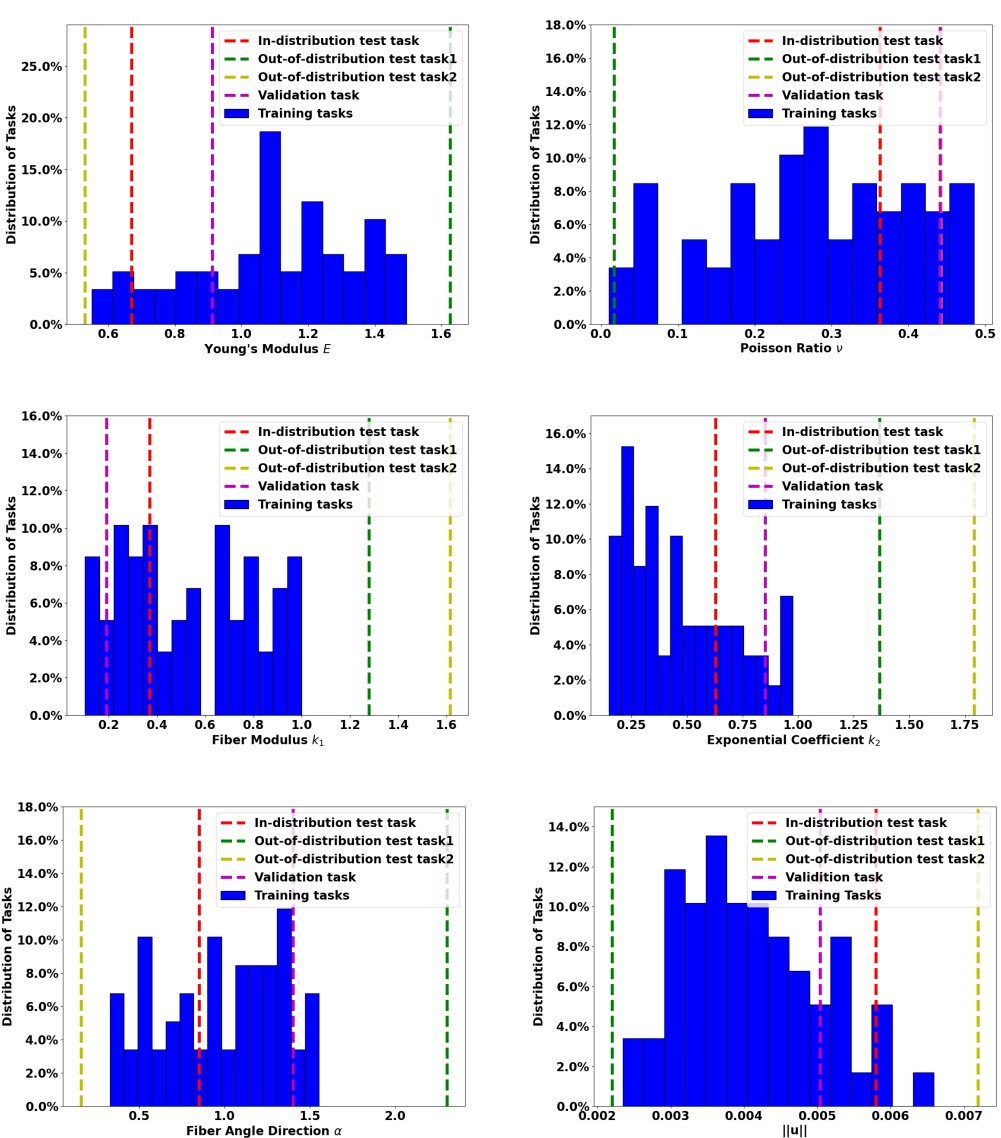

Figure 5: Distribution of physical parameters of different tasks, and the resultant magnitude of material response, $\|u^{\eta}(\mathbf{x})\|_{L^2(\Omega)}$, on an exemplar loading instance shown in Figure 4(b).

$\theta_P^\eta$ and the common parameters $\theta_I$ and $\theta_Q$ on all 59 tasks, with the context set of 500 samples on each task. After meta-train phase, we load $\theta_I$ and $\theta_Q$ and the averaged $\theta_P^\eta$ among all 59 tasks as initialization, then tune the hyperparameters based on the validation task. In particular, the 500 samples on the validation task is split into two parts: 300 samples are reserved for the purpose of training (as the context set) and the rest 200 samples are used for evaluation (as the target set). Then we train for the lift layer on the validation task, and tune the learning rate, the decay rate, and the weight decay parameter for different context set sizes ($N^{\text{test}}$), to minimize the loss on the target set. Based on the chosen hyperparameters, we perform the test on the test task by training for the lift layer on different numbers of samples on its context set, then evaluate and report the performance based on its target set. We repeat the procedure on the test task with selected hyperparemeters with different 5 random seeds, and calculate means and standard errors for the resultant test errors on target set.

**MAML&ANIL**: For MAML and ANIL, we use the same architecture as the base model, and also split the training tasks for the purpose of training and validation as in MetaP. During the meta-train phase, for each task we randomly split the available 500 samples to two sets: 250 samples in the support set used for inner loop updates, and the rest in the target set for outer loop updates. During the inner loop update, we train for the task-wise parameter with one epoch, following the standard settings of MAML and ANIL (Finn et al., 2017; Raghu et al., 2019). Then, the model hyperparameters, including the learning rate, weight decay, decay rate, and inner loop learning rate, are tuned. In the meta-test phase, we load the initial parameter and train for all parameters (in MAML) or the last-layer parameters (in ANIL) until the optimization algorithm converges. Similar as in MetaP, we first tune the hyperparameters on the validation task, then evaluate the performance on the test task.

## B.2 EXAMPLE 2: MECHNICAL MNIST

### B.2.1 DATA SETTINGS

Mechanical MNIST is a benchmark dataset of heterogeneous material undergoing large deformation, modeld by the Neo-Hookean material with a varying modulus converted from the MNIST bitmap images (Lejeune, 2020). In this example, we randomly select 102 specimens corresponding to the hand-written number "1". On each specimen, we have 32 loading/response data pairs on a structured 27 by 27 grid, under the uniaxial extension, shear, equibiaxial extension, and confined compression load scenarios, respectively. All 102 specimens are splitted into three groups: 100 specimens for the purpose of training in the meta-train stage, 1 specimen for validation, and 1 specimen for test. On the validation and test tasks, we reserve a target set consisting of 20 data pairs for the purpose of evaluation, then use the rest as the context set.

### B.2.2 ALGORITHM SETTINGS

**Base model**: As the base model for all algorithms, we construct two IFNO architectures, for the prediction of $u_x$ and $u_y$, the displacement fields in the $x$- and $y$-directions, respectively. On each architecture, the input loading field instance $\mathbf{g}(\mathbf{x}) \in \mathcal{A}$ is mapped to a higher dimensional representation via a lifting layer $\mathcal{P}[\mathbf{g}](\mathbf{x})$ parameterized as a 1-layer feed forward linear layer with width (4,64). Then for the iterative layer in equation 1, we set the number of truncated Fourier mode as 13, and parameterize the local linear transformation parameter, $W$, as a 1-layer feed forward network with width (64,64). In the projection layer, a 2-layer feed forward network with width (64,128,1) is employed. In this example we also apply the shallow-to-deep technique to accelerate the training, and set the layer depth as $L = 8$.

**MetaP**: During the meta-train phase, we train for the task-wise parameters $\theta_P^\eta$ and the common parameters $\theta_I$ and $\theta_Q$ on all 100 training tasks, with the context set of 32 samples on each task. After the meta-train phase, we load $\theta_I$ and $\theta_Q$ and the averaged $\theta_P^\eta$ among all 100 tasks as initialization, then train for $\theta_P$ on the validation task. In particular, the 32 samples on the validation task is split into two parts: 12 samples are reserved for the purpose of training (as the context set) and the rest 20 samples are used for the purpose of evaluation (as the target set). Then we train for the lift layer on the validation task, and tune the learning rate, the decay rate, and the weight decay parameter for different context set sizes ($N^{\text{test}}$), to minimize the loss on the target set. Based on the chosen hyperparameters, we perform the meta-test phase on the test task by training for the lift layer on

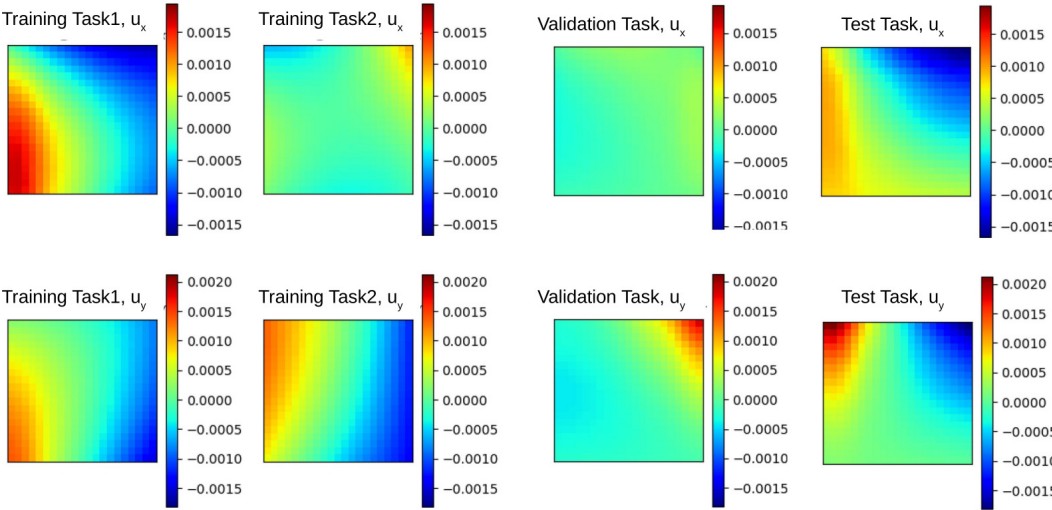

Figure 6: Visualization of the processed dataset in example 3: learning the biological tissue responses. Subject to the same loading instance, different columns show the corresponding ground-truth solutions $\mathbf{u}^{\eta}(\mathbf{x})$ from different tasks, showing the solution diversity across different tasks due to the change of underlying hidden material parameter field.

different numbers of samples on its context set, then evaluate and report the performance based on its target set.

**MAML&ANIL**: For MAML and ANIL, we use the same architecture as the base model, and also split the training tasks for the purpose of training and validation as in MetaP. During the meta-train phase, for each task we randomly split the available 32 samples to two sets: 16 samples in the support set used for inner loop updates, and the rest in the target set for outer loop updates. During the inner loop update, we also follow the standard settings of MAML and ANIL (Finn et al., 2017; Raghu et al., 2019), and tune the hyperparameters following the same procedure as elaborated above for Example 1.

### B.3 EXAMPLE 3: EXPERIMENTAL MEASUREMENTS ON BIOLOGICAL TISSUES

### B.3.1 DATA GENERATION

We now briefly provide the data generation procedure for the tricuspid valve anterior leaflet (TVAL) response modeling example. In this problem, the constitutive equations and material microstructure are both unknown, and the dataset has unavoidable measurement noise. To generate the data, we firstly followed the established biaxial testing procedure, including acquisition of a healthy porcine heart and retrieval of the TVAL Ross et al. (2019); Laurence et al. (2019). Then, we sectioned the leaflet tissue and applied a speckling pattern to the tissue surface using an airbrush and black paint Zhang & Arola (2004); Lionello & Cristofolini (2014); Palanca et al. (2016). The painted specimen was then mounted to a biaxial testing device (BioTester, CellScale, Waterloo, ON, Canada). To generate samples for each specimen, we performed 7 protocols of displacement-controlled testing to target various biaxial stresses: $P_{11} : P_{22} = \{1 : 1, 1 : 0.66, 1 : 0.33, 0.66 : 1, 0.33 : 1, 0.05 : 1, 1 : 0.1\}$. Here, $P_{11}$ and $P_{22}$ denote the first Piola-Kirchhoff stresses in the $x$- and $y$-directions, respectively. Each stress ratio was performed for three loading/unloading cycles. Throughout the test, images of the specimen were captured by a CCD camera, and the load cell readings and actuator displacements were recorded at 5 Hz. After testing, the acquired images were analyzed using the digital image correlation (DIC) module of the BioTester's software. The pixel coordinate locations of the DIC-tracked grid were then exported and extrapolated to a 21 by 21 uniform grid.

In this example, we have the DIC measurements on 14 specimens, with 500 data pairs of loadings and material responses from the 7 protocols on each specimen. These specimens are divided into three groups: 12 for the purpose of meta-train, 1 for validation, and 1 for test. To demonstrate the

diversity of these specimens due to the material heterogeneity in biological tissues, in Figure 6 we plot the processed displacement field of two exemplar training specimens and the validation and test specimens.

### B.3.2 ALGORITHM SETTINGS

**Base model**: As the base model, we first construct the lifting layer as a 1-layer feed forward linear layer with width (4,16). Then for the iterative layer in we keep 8 truncated Fourier modes and parameterize the local linear transformation parameter, $W$, a 1-layer feed forward network with width (16,16). In the projection layer, a 2-layer feed forward network with width (16,64,1) is employed. We construct two 4-layer IFNO architectures, for the prediction of $u_x$ and $u_y$, the displacement fields in the $x$- and $y$-directions, respectively.

**MetaP**: During the meta-train phase, we train for the task-wise parameters $\theta_P^\eta$ and the common parameters $\theta_I$ and $\theta_Q$ on all 12 tasks, with the context set of 500 samples on each task. After meta-train phase, we load $\theta_I$ and $\theta_Q$ and the averaged $\theta_P^\eta$ among all 12 tasks as initialization, then tune the hyperparameters based on the validation task. In particular, the 500 samples on the validation task is splited into two parts: 300 samples are reserved for the purpose of training (as the context set) and the rest 200 samples are used for evaluation (as the target set). Based on the chosen hyperparameters, we perform the test on the test task by training for the lift layer on different numbers of samples on its context set, then evaluate and report the performance based on its target set.

**MAML&ANIL**: For MAML and ANIL, we use the same architecture as base model, and also split the training tasks for the purpose of training and validation as in MetaP. During the meta-train phase, for each task we randomly split the available 500 samples to two sets: 250 samples in the support set used for inner loop updates, and the rest in the target set for outer loop updates. During the inner loop update, we train for the task-wise parameter with one epoch, following the standard settings of MAML and ANIL (Finn et al., 2017; Raghu et al., 2019).

