# OpenReview forum: "MetaP: How to Transfer Your Knowledge on Learning Hidden Physics"
_ICLR.cc/2023/Conference — Submitted to ICLR 2023_

### Official Review · Reviewer_evBd · 2022-10-21

**Confidence:** 2
**Correctness:** 3
**Technical Novelty And Significance:** 2
**Empirical Novelty And Significance:** 3
**Recommendation:** 5

**Clarity, Quality, Novelty And Reproducibility:**

The clarity of the paper could be improved a lot. In particular:
* Section 2 is very verbose and could be made much more concise
* The formal problem setting which is introduced in the intro with formulas (i.e. Eq. 1) is not very clear since relevant concepts have not been introduced yet at that point. For instance, it is not clear at all how $\mathbf{b}$ interacts with the rest of the PDE model. While the intro already talks about the solution of Eq. 7, the formula is only introduced 4 pages later. I think it would be more pedagogical to first talk about the ‘normal’ neural operators and then introduce the multi-task structure later.
* The notation is heavy and not very well explained. Since mathematical symbols often re-appears 2 pages later again without any reminders of their meaning, I found myself often scrolling back to find out how each symbol was defined. For instance, it would help the reading flow a lot if you would write ‘a function from the space of property fields B to the loading fields A’  instead of ‘a function from B to A’
* A lot of the formalization and explanation in the paper is very specific to mechanical engineering & physics. This makes the paper not very accessible to the broader machine-learning audience at ICLR.
* Assumptions 1 & 2 are not motivated much. If you make assumptions to prove a theorem you should explain why they are necessary and when they hold in practice.

The paper is novel in the sense that it is the first work I know of that does meta-learning with neural operators. The proposed approach itself is a straightforward combination of gradient-based meta-learning and Fourier neural operators. I do not find the universal approximation result very useful or enlightening. From how it is presented, it seems like the authors just assumed whatever they needed so that theory of Lu et al. (2019, 2021) goes through.

While the experiments include relevant baselines and appropriate evaluation problems, there are not enough details to reproduce the experiments. More critically, it seems like the experiments have not been repeated with multiple seeds, and no standard errors are reported. This casts serious doubts about the experimental methodology and the reliability of the presented experiment results.



**Strength And Weaknesses:**

Strengths:
* Relevant problem with clear application
* Real-world experiments
* The first meta-learning approach based on neural operators

Weaknesses:
* The presentation of the work could be improved a lot
* No standard errors are reported in the experiments. Were experiments even repeated with multiple seeds
* Not enough experiment details to reproduce experiments
* Assumptions in Section 3.3. are not motivated

Minor typos:
* 3rd paragraph on page 3: ‘neural operators requires’ -> ‘neural operators require’
* Page 4, ‘we denote the set trainable parameters’ ->  ‘we denote the set of trainable parameters’


**Summary Of The Paper:**

The paper proposes a MAML-like approach for neural operators aimed at transferring learning across slightly varying physical systems. It builds on recent work on implicit Fourier neural operators and proposes a meta-learning method that uses MAML for the (first) lifting layer, while the remaining layers (including the Fourier layers) are meta-learned but not adapted at meta-test time. The proposed method is experimentally evaluated on simulated PDE data as well as a real-world physical tissue response setting.


**Summary Of The Review:**

Overall, I think that the studied problem is relevant and the proposed method has its merits. Unfortunately, it is not very well presented which makes the paper hard to read. In addition, there are serious shortcomings in the experimental methodology and documentation. At the moment, I do not think the paper is ready for publication. Since I consider the proposed method relevant, I encourage the authors to seriously re-work and improve the writing of the paper based on my comments above.

---

> ### Author Response · Authors · 2022-11-19
> **Thank you and response to reviewer evBd**
>
> **Presentation of the work**: We have improved the writing of the paper based on the reviewer's valuable comments.
>
> **Repeated experiments**: We thank the reviewer's kind suggestion. We have repeated each experiment for 5 times, and reported the averaged errors and their standard errors in the revised manuscript.
>
> **Parameters and details of experiments**: We thank the reviewer's suggestion, and have added the parameters and details of each experiment in Appendix B. We will also release all codes and the processed datasets on Github once the manuscript is accepted, for readers to reproduce the experimental results.
>
> **Motivations of assumptions 1&2 and Novelty**: Assumptions 1&2 are motivated from the fixed-point iterative method -- the basic numerical solver for nonlinear PDE problems. Intuitively speaking, these two assumptions imply that there exists a converging numerical solver for the common PDE system in equation (8). Hence, these two assumptions are actually motivated from numerical PDE solving problems and are required conditions of them.
>
> In addition, we would also like to point out that we do not use  theoretical results from Lu et al. (2019, 2021). They focus on a completely different neural operator (DeepONet), and does not apply to the integral neural operator used in this work.

---

### Official Review · Reviewer_GCvR · 2022-10-23

**Confidence:** 3
**Correctness:** 3
**Technical Novelty And Significance:** 2
**Empirical Novelty And Significance:** 3
**Recommendation:** 5

**Clarity, Quality, Novelty And Reproducibility:**

The paper is clearly written and the proposed model, experiments and results indicate a good quality paper. Unable to verify  / reproduce results due to non access to the source code.

**Strength And Weaknesses:**

## Strengths:

**[Novel Few-Shot Operator Transfer Learning]**: Authors address an important problem of few-shot learning in the context of knowledge transfer across PDE domains. Specifically, the proposed MetaP, MetaP+ methods learn to effectively transfer knowledge with only a few instances of data from the target task of interest.


**[Demonstration on Synthetic and Real-world Data]**: The proposed framework not only demonstrates good results on a set of synthetic PDE based domains but also on real-world task of learning to estimate mechanical response of biological tissue specimen from displacement tracking measurements.


**[Effective Few-shot Performance]**: The MetaP, MetaP+ models demonstrate very effective few-shot generalization performance with as little as 2 instances required to achieve <5% prediction errors on synthetic data with similar results also on real-world datasets.


**[Baseline Comparison]**: Authors compare with popular meta learning models MAML [2], ANIL [3] and a version of the base model [4] without meta-learning as ablation baselines. The proposed models outperform all baselines under few-shot learning settings.



## Weaknesses:

**[Some Baselines Missing]**: A related few-shot operator learning baseline model [5] based on one-shot-learning with deep operator networks is missing.


**[Out-of-Distribution Results Require Further Clarity]**: Some of the out-of-distribution based experiments seem to produce results that are better than in-distribution test results. Without qualitative visualizations of training and testing regimes (out-of-distribution case), it is hard to comprehend the degree of challenge posed by out of distribution data regime.


**[Limited Novelty]**: Although the problem is well motivated and results of the proposed pipeline are somewhat convincing, the proposed framework itself is not significantly novel. Specifically, the sole contribution of the authors may be considered to be the insight of transfer learning by fine-tuning the “lifting layer” in the neural operator (backed by theoretical and quantitative results) instead of the last layer as in traditional transfer-learning paradigms. Other parts of the model (I.e., based model , meta-learning framework) are directly employed from previously published work.

**[Typos]**: Paper has a few typos (e.g., Eq. 7 is referenced in multiple places in reference to the solution operator `G` , but Eq. 7 contains no such symbol.

## Questions for Authors:

How has model architectures (number of layers, activations) selected? What is the validation set size and how was it selected?


Please provide an explanation in terms of a performance breakdown for why the out-of-distribution test error of the proposed MetaP model is lower than in-distribution test error? This seems surprising considering the challenge of the task?


Could you please provide justification about the difference in the domain characteristics in the training and testing regimes for out-of-distribution tasks? For example, please comment on what about the test regimes makes (e.g., more stiff) make them a true test of out-of-distribution performance?


How were out-of-distribution ranges selected for synthetic datasets in section 4.1?



## References:

1. Li Z, Kovachki N, Azizzadenesheli K, Liu B, Bhattacharya K, Stuart A, Anandkumar A. Fourier neural operator for parametric partial differential equations. arXiv preprint arXiv:2010.08895. 2020 Oct 18.


2. Finn C, Abbeel P, Levine S. Model-agnostic meta-learning for fast adaptation of deep networks. In International conference on machine learning 2017 Jul 17 (pp. 1126-1135). PMLR.


3. Raghu A, Raghu M, Bengio S, Vinyals O. Rapid learning or feature reuse? towards understanding the effectiveness of maml. arXiv preprint arXiv:1909.09157. 2019 Sep 19.


4. Li Z, Kovachki N, Azizzadenesheli K, Liu B, Bhattacharya K, Stuart A, Anandkumar A. Neural operator: Graph kernel network for partial differential equations. arXiv preprint arXiv:2003.03485. 2020 Mar 7.


5. Lu L, He H, Kasimbeg P, Ranade R, Pathak J. One-shot learning for solution operators of partial differential equations. arXiv preprint arXiv:2104.05512. 2021 Apr 6.

**Summary Of The Paper:**

Authors propose a (somewhat) novel method that combines meta-learning with a popular operator learning model (based on the Fourier-Neural Operator learning paradigm) and develop a few-shot learning framework to learn PDE based domains. Additionally, authors also introduce a novel modification to the few-shot inner-loop model update wherein the updates are only localized to the “lifting” layers in the FNO based operator framework.

**Summary Of The Review:**

The paper proposes a novel few-shot learning based neural network based operator learning framework. Although the results are convincing, the proposed model is significantly based on previously proposed research efforts and the main contribution of the paper is their insight about fine-tuning the "lifting layer" instead of the final layer as in traditional transfer learning paradigm. Overall, the novelty of the proposed method is somewhat lacking but the incrementally novel proposed model does produce effective few-shot transfer learning results.

---

> ### Author Response · Authors · 2022-11-19
> **Thank you and response to reviewer GCvR**
>
> **Missing baselines**: We thank the reviewer's valuable suggestion. However, we would like to point out that in [1] the one-shot-learning relies on the assumption that one can choose the sample input function, such that this one sample can ''induce a more diverse training dataset'' and can be partitioned as many local samples. Such a setting is generally infeasible in the synthetic and real-world problems discussed on our paper, where the loading patterns as well as measurement resolutions are limited. We have added reference [1] to the revised manuscript.
>
> [1]Lu L, He H, Kasimbeg P, Ranade R, Pathak J. One-shot learning for solution operators of partial differential equations. arXiv preprint arXiv:2104.05512. 2021 Apr 6.
>
> **Out of distribution (OOD) task and the data distributions**: We thank the reviewer's valuable suggestion, and have provided the distribution of physical parameters in training and testing regimes as well as the $L^2$ norm of solutions in the Appendix B (see Figure 5) of the revised manuscript. These plots demonstrate that the material parameters in the OOD task, $\alpha$, $k_1$, $k_2$ and $E$, are not covered in the training regime, and hence provide justifications about the difference in the domain characteristics in the training and testing regimes. Moreover, from the distribution of solutions, one can see from Figures 4-5 in the revision that the solutions in the original OOD task (denoted as ''OOD task 1'') generally have smaller magnitudes, due to the fact that the larger physical parameter in this task induces smaller deformation of the material. Therefore, the solution operator in  ''OOD task 1'' lies closer to a linear regime, and makes the solution operator learning task easier. To clarify this problem and further demonstrate the performance of MetaP in more challenging tasks, we have added an additional OOD task (denoted as ''OOD task 2'') which corresponds to softer materials and hence induces larger deformation of the specimen. One can see that the accuracy in OOD task 2 is worse than the in-distribution (ID) task, while the errors in all three tasks (two OOD tasks and the ID task) have similar trends. This result verifies the fact that the learnt operator generalizes well to PDEs with different parameter sets, since the accuracy are mostly associated with the level of nonlinearity of each task.
>
> **Out-of-distribution (OOD) ranges**: The physical parameter for OOD task 1 was chosen to produce a ''stiffer'' specimen. As discussed above, we have added an additional OOD task, which corresponds to a ''softer'' specimen, so as to provide a more complete assessment for the proposed MetaP in different test scenarios.
>
>
> **Parameters and details of experiments**: We thank the reviewer's suggestion, and have added the parameters and details of each experiment in Appendix B.
>
> **Novelty of the proposed MetaP**: To the authors' best knowledge (and as pointed out by Reviewer evBd), this is the first work combining meta-learning with neural operators with provable guarantees. Moreover, we have provided new theoretical justifications, which suggested the novel architecture of adapting the first layer. This is a new finding different from standard meta-learning approaches (such as in ANIL).

---

### Official Review · Reviewer_koC6 · 2022-10-25

**Confidence:** 3
**Correctness:** 4
**Technical Novelty And Significance:** 3
**Empirical Novelty And Significance:** 3
**Recommendation:** 6

**Clarity, Quality, Novelty And Reproducibility:**

The writing of this paper is clear and decent. The idea is novel and interesting. It would be better if the author could release the source code for readers to reproduce the experimental results.

**Strength And Weaknesses:**

* Strength
1. The idea of transferring knowledge between neural operators is interesting and novel
2. It adapted the first layer of the neural operator model to capture hidden parameter field in contrast to the final-layer transfer in existing meta-learning methods
3. Conduct extensive experiments on benchmark and real-world datasets to verify the effectiveness of the proposed approach

* Limitations
1. It only adopted Fourier Neural Operator to show the effectiveness of the method, and it would be better to study more neural operators to verify this idea.
2. Please repeat the experiments multiple times and then report the average error and standard deviation.
3. A couple of grammatical issues. For example, "As an motivating example," on page 1; and "ANIL In " on page 5, I am curious if a reference is missing?
4. Questions:
[1] In fig 2, why the accuracy of OOD task is better than ID task? Please provide possible explanation on it in more detail.
[2] I am curious where could we access the real-world dataset: biological tissue specimens from DIC displacement?
[3] I am wondering if the proposed method could learn multi-tasks from different PDEs rather than one common PDE with different physical parameter?

**Summary Of The Paper:**

This work developed a novel meta-learnt method to transfer learning knowledge between neural operators. Different from typical
final-layer transfer in existing meta-learning methods, this paper adapted the first layer of the neural operator model to capture hidden
parameter field. Evaluation on both synthetic and real world datasets demonstrate the efficacy of the proposed approach.

**Summary Of The Review:**

It is interesting and novel to transfer the knowledge between neural operators. It would be much better to explore more neural operators to further verify the effectiveness of the proposed method.

---

> ### Author Response · Authors · 2022-11-19
> **Thank you and response to reviewer koC6**
>
> **Extension to more neural operators**: In all three examples considered in the paper, measurements are provided on structured grid points. Hence, we choose to employ FNO as the base neural operator model due to its efficacy and accuracy in handling structured grids. More general neural operator models will be considered in future work for problems with unstructured grids.
>
> **Repeated experiments**: We thank the reviewer's kind suggestion. We have repeated each experiment for 5 times, and reported the averaged errors and their standard errors in the revised manuscript.
>
> **The performance of OOD tasks**: As shown in Figures 4-5 of the revised manuscript, the solutions in the original OOD task (denoted as ''OOD task 1'') generally have smaller magnitudes, due to the fact that the larger physical parameter, $E$, in this task induces a stiffer material and correspondingly smaller deformation of the material. Therefore, the solution operator in  ''OOD task 1'' lies closer to a linear regime, and makes the solution operator learning task easier. To clarify this problem and further demonstrate the performance of MetaP in more challenging tasks, we have added an additional OOD task (denoted as ''OOD task 2'') which corresponds to softer materials and hence induces larger deformation of the material. One can see that the accuracy in OOD task 2 is worse than the ID task, while the errors in all three tasks (two OOD tasks and the ID task) have similar trends. This result verifies the fact that the learnt operator generalizes well to PDEs with different parameter sets, since the accuracy are mostly associated with the level of nonlinearity of each task.
>
> **Accessibility of the tissue dataset**: The biological tissue specimens measurements were produced in our lab. We will share all codes and the processed datasets on github once the manuscript is accepted.
>
> **Multi-tasks from different PDEs**: The current theory only demonstrates the universal approximation property for the case of learning one common PDE with different physical parameters. It will be an interesting direction to study on learning solution operators from different PDEs.

---

> > ### Comment · Reviewer_koC6 · 2022-11-20
> > **Thank you for your rebuttal**
> >
> > Thanks a lot for your rebuttal. My concern is that the proposed method is only limited to FNO, but not for many the physics-informed neural networks.

---

> > > ### Author Response · Authors · 2022-11-20
> > > **Thanks again and further response to reviewer koC6**
> > >
> > > We thank the reviewer again for the followup question.
> > >
> > > **Application to PINNs**: If the reviewer's concern is on physics-informed neural networks (PINNs) [1], we would like to point out that it is very different from the neural operators considered in our work. PINNs generally requires at least a known parameterized partial differential equation (PDE) [1-2], which does not apply to the ''learning the hidden physics'' scenario discussed in our paper. This is partially the reason why neural operators [3-4], mainly the integral neural operators [3] discussed in this paper and the DeepONets [4], were introduced, since they don't require any known parameterized partial differential equation form and are generalizability to different input instances.
> > >
> > > **Application to other neural operators**: Although we only implement it on FNO,  the main idea of our method can be immediately applied to meta learning with any neural operators. For integral neural operators discussed in [3], our method directly applies by using task-wise parameters on the lifting layer. For DeepONet [4] which uses branch/trunk net instead of lifting/iterative/projection layers, although the architecture is totally different, the insights from our theoretical analysis (in particular, equation (9)) still apply, and hence our main insight of ''designing task-wise parameters as the first layer after inputs, to reflect the changes on hidden physical parameters'' works as well. By setting the first layer of branch net as the task-wise parameter, the changes of hidden physical parameter fields will be encoded as the $F_1[b]$ term in our equation (9), and our analysis can be adapted with DeepONet as the base model following [4-5]. We will add such a discussion.
> > >
> > > [1] M. Raissi, P. Perdikaris, G. E. Karniadakis, Physics-informed neural networks: A deep learning framework for solving forward and inverse problems involving nonlinear partial differential equations, Journal of Computational Physics 378 (2019) 686–707.
> > >
> > > [2] G. E. Karniadakis, I. G. Kevrekidis, L. Lu, P. Perdikaris, S. Wang, L. Yang, Physics-informed machine learning, Nature Reviews Physics 3 (6) (2021) 422–440.
> > >
> > > [3] N. Kovachki, Z. Li, B. Liu, K. Azizzadenesheli, K. Bhattacharya, A. Stuart, A. Anandkumar, Neural operator: Learning maps between function spaces, arXiv preprint arXiv:2108.08481 (2021).
> > >
> > > [4] L. Lu, P. Jin, G. Pang, Z. Zhang, G. E. Karniadakis, Learning nonlinear operators via DeepONet based on the universal approximation theorem of operators, Nature Machine Intelligence 3 (3) (2021) 218–229.
> > >
> > > [5] T. Chen, H. Chen, Universal approximation to nonlinear operators by neural networks with arbitrary activation functions and its application to dynamical systems, IEEE Transactions on Neural Networks 6 (4) (1995) 911–917.

---

### Official Review · Reviewer_dEyR · 2022-10-25

**Confidence:** 4
**Correctness:** 3
**Technical Novelty And Significance:** 2
**Empirical Novelty And Significance:** 2
**Recommendation:** 5

**Clarity, Quality, Novelty And Reproducibility:**

As mentioned above, the paper describes a meta-learning methodology without using the standard terminology for meta-learning. For instance, what is exactly N_test (is it context set size) and what is it counterpart during training is not clear. Because of this, the clarity of the paper is affected.

Novelty of the paper as mentioned above is limited. It involves apply MAML to neural operators, and the main contribution seems to be at identifying that in neural operator the first layers should be adapted. It lacks discussion and comparison with related works that deal with approximating PDE-solvers, especially those with a meta-learning or generalization focus.

**Strength And Weaknesses:**

Strengths:

To be able to learn a neural operator for complex PDE-governed physics without varying parameters is important. The use of meta-learning is intuitive.

The experiments attempted various types of dataset beyond benchmark data, which is appreciated.



Weakness:

- A major concern is regarding the baseline. The primary baselines includes a "single" model trained on the meta-training data with multiple tasks, and three variants of meta-formulation. The comparison with the "single" model however is not fair. All the meta-models are further optimized to the context data (referred to as test data in the paper) at meta-test time; this was not accessible to the "single model". Two stronger baselines need to be added. One is the "single" model fine-tuned to the same meta-test support data (used to optimize the model for all the other meta-models during meta-testing). The other is the "single" model trained to the "single" task used in the meta-training set (and fine-tuned to the meta-test task depending on how different the tasks are).


- Another major concern is regarding the comparison to alternative models used to approximate PDE-based physics -- there is currently a large set of literature that does this; some of them are designed to generalize across different parameter settings (examples below). These works should be included for comparisons.

Yin et al. LEADS: Learning Dynamical Systems that Generalize Across Environments
Wang et al. Meta-Learning Dynamics Forecasting Using Task Inference


- The rationale for the choice of meta-learning formulations is not clear. There are many different frameworks for meta-learning, some of them do not require test-time optimization. Why this formulation versus alternatives?


- The difference between ANIL and MetaLast is not clear -- both does inner optimization to the projection layer? Please clarify.

- Intuitively, since MAML optimizes all parameters of the base model, it is not clear why it would be sub-optimal then optimizing only the first layer (except for computational overhead)? Comments on this would be appreciated.

- The wording of this paper is in general somewhat confusing, since it did not utilize the standard terminology of context/target samples for meta-learning. Assuming that N_test is referring to the context set size (i.e., size of meta-test samples used for inner loop adaptation at meta-test time), what is the corresponding context set size during meta-training? Is it the same as what is used at test time? Fig 2 showed how the test error changes as N_test changes. Is this N_test only referring to the context set size used during meta_test, or is the same parameter changing during meta_testing. In all experiments, N_test is given but not the size used during meta_training. This needs to be clarified. If N_test is different from the context set size used in meta-training, the effect of the latter would be more interesting to see in ablation studies.



- In synthetic experiments, it'd be good to more specifically describe exactly which parameters are used for meta-training, and which for meta-testing. i.e., it is good to have a sense of the distance between training and test task for in-distribution tasks.

- In real-world data experiments, it was mentioned that "we have 500 available data pairs" for each specimen, but it was then stated that MetaP was trained based on but N test = 500 samples and then evaluated on another 200 samples? If only 500 pairs are available, how can we use all of them as the context set? Please clarify


**Summary Of The Paper:**

This paper presents an approach to use MAML-based meta learning to learn a neural operator that approximates complex PDE-based equations with different physical parameters. Instead of optimizing all parameters of the base model during inner optimization (like vanilla MAML) or only the last layer (like ANIL), this paper's main contribution was to show that the first layer of the neural operator should be optimized. Experiments were conducted both on synthetic data, benchmark data, and a real-world data that describes mechanical response of biological tissue specimens from DIC displacement tracking. Experiments were mainly concerned with baselines using the neural operator without the presented meta-learning formulati

**Summary Of The Review:**

This paper applies a MAML variant to a neural operator for learning to approximate PDE-based physics for various parameters. The motivation is intuitive, and the point on having to adapt the first layer of the neural operator is interesting. Overall, however, this paper can be improved both in its novelty in relation to existing works, and stronger baseline/comparison methods.

---

> ### Author Response · Authors · 2022-11-19
> **Thank you and response to reviewer dEyR -- part 1**
>
> **Alternative meta-learnt models for PDE-based physics**: We thank the reviewer for point out the possible relevant work [1-2] on meta-learning of ODE/PDE-based physics. However, we would like to point out that these two papers only focused on dynamics forecasting, by learning the temporal evolution information directly (in [1]) or learning time-invariant features (in [2]). These two frameworks are not readily applicable to the general time-independent PDE solving tasks considered in our work. In fact, the known temporal structure, $\dfrac{d x}{dt}=f(x)$, has added additional information to the solving tasks, and the provided initial conditions can also be seen as prior knowledges on the predictions for the next time step. Therefore, learning the temporal evolution of ODE/PDE is a very different (and sometimes easier) task comparing with the general time-independent PDE solving task considered in our proposed MetaP. We have added the above discussion to Section 2.2 as well as references [1-2] to the references in the revised manuscript.
>
> [1] Yin, Yuan, et al. ''LEADS: Learning dynamical systems that generalize across environments.'' Advances in Neural Information Processing Systems 34 (2021): 7561-7573.
>
> [2] Wang, Rui, Robin Walters, and Rose Yu. ''Meta-learning dynamics forecasting using task inference.'' Advances in Neural Information Processing Systems 34 (2022).
>
>
> **Novelty of the proposed MetaP**: To the authors' best knowledge (and as pointed out by Reviewer evBd), this is the first work combining meta-learning with neural operators with provable guarantees. Moreover, besides suggesting the novel architecture of adapting the first layer with theoretical justifications, which is a new finding different from standard approaches (such as in ANIL), we have for the first time considered the transferability of multiple neural network-based PDE solution operators for the general time-independent PDEs.
>
>
> **Fair comparison with the ''single'' model**: We thank the reviewer's kind suggestion, and would like to point out that the ''single'' model  in the original paper was indeed the first suggested model baseline (i.e., fine-tuned to the same meta-test context data) suggested by the reviewer. Moreover, per suggestion, we have added an additional ''single'' model baseline (denoted as ''single+'' in the revised paper), which was trained to all context data in the meta-training set and fine-tuned to the meta-test task in the revision. Empirical experiments show that the ''single+'' model baseline outperforms the original ''single'' model, while the proposed MetaP approach still has a better performance in the small data regime.
>
>
>
> **Choice of meta-learning formulation**: In the hidden physics learning problem considered in this paper, test-time optimization would be an necessity. Firstly, we would like to highlight the fact that in our learning tasks the environment parameter, $b(x)$, is assumed to be hidden, so as to reflect the requirements of material modeling from experimental observations. $ b(x)$ can vary greatly from task to task. Therefore, when adapting the model to a new and unseen test task, additional observations are required to implicitly infer the change of environment parameter. Moreover, when the hidden physical parameters are altered, the underlining PDE as well as its solution operator are changed drastically, and therefore previous models (without adaption) are generally not applicable. To demonstrate the diversity of different tasks, we plot the exemplar material responses under the same loading instances for the synthetic dataset and tissue dataset in Figures 4 and 6 in the revision, respectively. One can see that the material responses can be drastically different in different tasks, due to the diversity of the hidden environment parameters.
>
>
> **Differences between ANIL and MetaLast**: These two models differ at the meta-train stage. In ANIL,  task-wise parameters were approximated with the inner loop update formulation (see equation (7) in the main text), using a subset of context samples on each training task (denoted as the support set, $S^\eta$ in our main text). On the other hand, MetaLast does not have an inner loop update. In MetaLast, we optimize  task-wise parameters with a gradient-based optimizer based on all context samples of each training task. At the meta-test stage, both models update the last layer using all context set samples.
>
>
> **Why MAML is suboptimal**: In the meta-test stage of MAML, all parameters are fine-tuned from the common initial parameter, $\tilde{\theta}$, based on few labeled samples. Therefore, comparing with MetaP, MAML has a much larger number of trainable parameters, which may more likely be susceptible to overfit in a small sample regime.

---

> > ### Comment · Reviewer_dEyR · 2022-11-29
> > **Thanks for the response**
> >
> > I want to thank the authors for the response.
> >
> > The terminology of the manuscript remains somewhat confusing. In the appendix (using B.3.2 as an example), it writes " During the meta-train phase, we train for the task-wise parameters and the common parameters all 12 tasks, with the context set of 500 samples on each task". Are 500 examples the context set used to optimize the task-wise parameters in the inner loop? If yes, what are the size of the target set used to optimize the common parameters during meta-training? This parameter was not mentioned throughout the manuscript.
> >
> > Regarding the "single" baseline -- I was referring to two baselines: one trained on meta-training data of all tasks and then fine-tuned to per-task context set used in meta-testing (this seems to be the single+), and the other trained on meta-training data of a single task and either directly applied to context set used for the same task or fined-tuned to context set used for a new task at meta-testing data. It seems that the "single" baseline presented is only trained on context set used during meta-testing (did not use meta-training data). This needs to be clarified.
> >
> > The effect of the size of the context set at training time as mentioned in my original review would be important to see, which is currently fixed (and pending confusion to be clarified in the first question above) and its effect not investigated.
> >
> > With details of data splitting added in appendix, I noticed that all experiments only considered one left-out task for meta-testing. Unless this is repeated in a cross-validation type of experiments, there is concern that leaving out only one task for testing would not give comprehensive understanding of the performance of the model. Even if this is repeated one-task-left-out a time, it'd seem to be more reasonable to test the method when multiple tasks are left out (which are standard in meta-learning literature).
> >
> > Overall I also shared the concern of other reviewers regarding the novelty and contribution of the presented work, which seems to be heavily tied to the first use of MAML for neural operators.

---

> > > ### Author Response · Authors · 2022-12-10
> > > **Thanks for the additional comment**
> > >
> > > We thank the reviewer for further detailed discussion, and have included new results and further clarification below.
> > >
> > > **Number of samples for inner/outer loop updates**: As our previous discussion on MetaLast, MetaP also does not use any inner loop update, so we used all context set samples to train all parameters. For those methods using an inner loop update, e.g., MAML and ANIL, we have included those information in the paragraph just below the sentence that the reviewer quoted in B.3.2 (and similarly in B.1.2 and B.2.2 for other two examples): ''250 samples in the support set used for inner loop updates, and the rest in the target set for outer loop updates.'' Here, we used the original terminology (support set and target set) employed in MAML and ANIL, and the union of these two sets is the context set.
> > >
> > > **Single baseline**:  We thank the reviewer for further clarification, and added this additional baseline in experiments (denoted as single-minus, ''single-''). As can be seen in https://ibb.co/QXgw4Nw, the single task suggested here has a similar performance as the tested single and single+ baselines, but worse than our proposed method.
> > >
> > > **Effect of the size of the context set**: Per the reviewer's request, we have added additional ablation study to the synthetic dataset (HGO), to test the performances of MetaP, MetaP+, MAML and ANIL with context sizes {50,100,200,500} on each training task. Here, MetaP and MetaP+ did not use any inner loop (all parameters from all training tasks are optimized together), and in MAML and ANIL we used half of the context set for inner loop updates (support set) and the other half for outer loop updates (target set). Results are attached in https://ibb.co/Cs8Q7Zx. One can see that with more context data, all methods have improved performance, but overall MetaP and MetaP+ still achieve the best results. Specifically, when using a small number of labelled data pair on the test task (X-axis with $N^{test}\leq 20$), MetaP and MetaP+ consistently achieve the best performance among all methods compared. when $N^{test} > 20$ , the fine-tune step over all parameters on small context sizes would help more than on larger context sizes, as indicated by the better performance of MAML and MetaP+ over ANIL and MetaP.  Overall, with larger $N^{test}$, MetaP+ performs the best.
> > >
> > > **Additional left-out tasks**:  Per the reviewer's request, we have added additional test tasks (5 test tasks in the synthetic data experiment and the benchmark data experiment, and 3 test tasks in the real-world experiment since the acquirement of specimens is challenging) to all experiments. As an exemplar, the results on the HGO dataset can be found at https://ibb.co/QXgw4Nw and https://ibb.co/s5GvbyQ (which correspond to the two figures in Figure 2 in the paper). The new results on 5 tasks show very similar trends as the previously reported result on one task in the paper, again confirming the promising result of our proposed method.
> > >
> > >
> > > **Novelty**: As we have pointed out to other reviewers, our architecture is not an extension of MAML, but a novel architecture change by adapting the first layer. This architecture change is motivated by our universal approximator analysis on parametric PDE solving tasks, and it is substantially different from existed popular meta-learning approaches such as MAML and ANIL. In fact, both MAML and ANIL rely on the adaptation of their last layer, as shown in [1-2], and this property makes them not suitable for PDE solving tasks.
> > >
> > > [1] Collins L, Mokhtari A, Oh S, Shakkottai S. ''MAML and ANIL provably learn representations.'' Proceedings of the 39 th International Conference on Machine Learning. 2022.
> > >
> > > [2] Raghu, Aniruddh, et al. ''Rapid Learning or Feature Reuse? Towards Understanding the Effectiveness of MAML.'' International Conference on Learning Representations. 2019.

---

> ### Author Response · Authors · 2022-11-19
> **Thank you and response to reviewer dEyR -- part 2**
>
> **Terminology for meta-learning**: We have been following the notations of ANIL [3], where they did not use any context/target set terminology. We thank the reviewer's kind suggestion, and have clarified the equivalence of the context/target set terminology (see the footnote 1 of page 2) as well as the detailed experimental settings for further clarification. In this paper, $N^{test}$ denotes the context set size in meta-test stage and it is different from the context set size used in meta-training. In each example, the context set during meta-training is chosen to be all labelled samples on each training task, which does not alter with the change of $N^{test}$. We have clarified the exact data split in Algorithm settings of Appendix B, for each algorithm in each experiment.
>
> [3] Raghu, Aniruddh, et al. ''Rapid Learning or Feature Reuse? Towards Understanding the Effectiveness of MAML.'' International Conference on Learning Representations. 2019.
>
> **Parameters and details of experiments**: We thank the reviewer's suggestion, and have added the parameters and details of each experiment in Appendix B.
>
> **Number of data pairs in real-world data experiments**: We will correct the typo on the $N^{test}$ size, which should be 300.

---

### Author Response · Authors · 2022-11-19
**Thank you**

We thank the reviewers for their valuable time, detailed comments, and constructive feedback. We have taken the time to address each issue in the revision, with changes highlighted in blue, and responded to the major questions of each reviewer.

Specifically, we have:

- improved and clarified many notation and presentations throughout the paper.

- added relevant baselines and discussed other baselines that are not readily applicable to our tasks.

- clarified the difference and novelty of our work and theoretical results,  in comparison to existing works. We also show the assumptions used are typical in most numerical PDE solving tasks.

- added substantial amount of experimental details, addressed the OOD task performance, and repeated the experiments 5 times per suggestion.

Hopefully, we have answered most, if not all, questions from the reviewers between the response below and the revised manuscript.

---

### Decision · Program_Chairs · 2023-01-20

**Decision:**

Reject

**Justification For Why Not Higher Score:**

There was a consensus among the reviewers that the paper needs a major revision, there are too many elements to modify.

**Justification For Why Not Lower Score:**

N/A

**Metareview: Summary, Strengths And Weaknesses:**

This paper presents a meta-learning approach to transfer knowledge from neural operators for PDE approximations in order to transfer knowledge to unseen physical systems. A part of the contribution is to show that transferring from the first layer of the neural operator model is better. The method is evaluated on synthetic datasets, MNIST, and real datasets not governed by PDE.

Strengths:
-idea interesting for an important problem
-good intuitive explanations
-experimental evaluation both on synthetic and real-world data

Weaknesses:
-the writing of the paper lacks clarity in some aspects
-the novelty has to be better justified with respect to other baselines
-some aspects of the experimental setup were not convincing

In the rebuttal, authors have provided additional results and precisions on their work.
During discussion, it has been recognized that the authors made an important effort to improve and precise their work, however there was a consensus among the reviewers to say the work still needs a global revision and that the elements provided are not sufficient for ICLR.
I propose then rejection.